# Reconstructing the functions of endosymbiotic Mollicutes in fungus-growing ants

Panagiotis Sapountzis*, Mariya Zhukova, Jonathan Z Shik, Morten Schiott, Jacobus J Boomsma*

Centre for Social Evolution, Department of Biology, University of Copenhagen, Copenhagen, Denmark

**Abstract** Mollicutes, a widespread class of bacteria associated with animals and plants, were recently identified as abundant abdominal endosymbionts in healthy workers of attine fungus-farming leaf-cutting ants. We obtained draft genomes of the two most common strains harbored by Panamanian fungus-growing ants. Reconstructions of their functional significance showed that they are independently acquired symbionts, most likely to decompose excess arginine consistent with the farmed fungal cultivars providing this nitrogen-rich amino-acid in variable quantities. Across the attine lineages, the relative abundances of the two Mollicutes strains are associated with the substrate types that foraging workers offer to fungus gardens. One of the symbionts is specific to the leaf-cutting ants and has special genomic machinery to catabolize citrate/glucose into acetate, which appears to deliver direct metabolic energy to the ant workers. Unlike other Mollicutes associated with insect hosts, both attine ant strains have complete phage-defense systems, underlining that they are actively maintained as mutualistic symbionts.
DOI: https://doi.org/10.7554/eLife.39209.001

*For correspondence:
sapountzis@bio.ku.dk (PS);
jjboomsma@bio.ku.dk (JJB)

**Competing interests:** The authors declare that no competing interests exist.

## Introduction

Bacterial endosymbionts, defined here as comprising both intra- and extra-cellular symbionts (*Bourtzis and Miller, 2006*), occur in all eukaryotic lineages and range from parasites to mutualists (*Bourtzis and Miller, 2006*; *Martin et al., 2017*). Their genomes tend to evolve faster than those of free-living bacteria (*Delaney et al., 2012*; *Moran et al., 1995*) and they often rely on recombination and horizontal gene transfer when their tissue localizations allow frequent DNA exchange with other bacteria, which tends to purge deleterious mutations when effective population sizes are small (*Naito and Pawlowska, 2016*; *Takeuchi et al., 2014*). Host-level selection can also induce radical changes in the gene content of endosymbionts (*Wernegreen, 2002*). When they are pathogens such changes can be adaptations to prevail against host defenses or competing bacteria (*Didelot et al., 2016*), as expected from arms races with Red-Queen dynamics (*Mallo et al., 2002*; *Paterson et al., 2010*). However, when symbionts are mutualists and provide nutritional services, they may become so tightly co-adapted to their hosts that they resemble organelles (*Douglas, 1996*; *Brinza et al., 2009*). In such cases natural selection is expected to have purged any genes that mediated functions that could be provided more productively by the hosts, a process that has been referred to as Black-Queen dynamics (*Morris et al., 2012*).

The increasing availability of sequenced genomes and accurate molecular phylogenies (e.g. *Leclercq et al., 2014*; *Gerth et al., 2014*) has allowed a number of intricate endosymbioses between bacteria and arthropods to be understood at functional metabolic levels well beyond qualitative assessments based on 16S ribosomal sequencing. Comparative genomics studies have detected gains and losses of genes or pathways when specialized endosymbionts co-evolve with

**eLife digest** Bacteria live inside the gut of most creatures. Some are harmful, some beneficial, and some have no clear effects. Studying the genetic material (the genome) of gut bacteria has revealed how they can improve the health, efficiency, and reproductive success of their hosts. For example, studies show that insects with low nutrient diets often benefit from gut bacteria that make vitamins or help them convert food into energy.

Panamanian leafcutter ants live in large colonies and farm fungus for food. They harvest leaves to feed their fungus farms and many are major crop pests in Latin America. How they evolved to be so successful is unclear. Recent studies have shown that huge numbers of bacteria called Mollicutes live in the leafcutter ants' guts. These bacteria do not make the ants sick, so they were thought to be somehow beneficial.

Now, Sapountzis et al. show that the two most common types of Mollicutes found in leafcutter ants evolved to make fungus farming more efficient. The complete genomes of two Mollicutes strains were analyzed and compared to the ones found in other insects. The results showed that both types of Mollicutes can turn excess quantities of the amino acid arginine into a nitrogen-rich fertilizer the ants deposit on their fungal gardens as feces. This helps the ants produce more food. One of the two types can also decompose citrate from plant sap and fruit juice consumed by the ants. This produces acetate, which supplements the ants' fungal diets and provides extra energy.

The unique energy-producing Mollicutes may explain why leafcutter ants evolved larger colonies and sustain higher levels of worker activity than other species of fungus-growing ants. The genome data also showed that both types of Mollicutes have costly defense systems to protect themselves against bacteria-destroying viruses. Many bacteria do not invest is such systems, but the Mollicutes may be able to afford them because their ant hosts provide them with plenty of food. This suggests that both the ants and the Mollicutes benefit from their symbiotic relationship.

DOI: https://doi.org/10.7554/eLife.39209.002

arthropod hosts (*Wernegreen, 2002*; *Didelot et al., 2016*; *Moran et al., 2008*), and have shown that bacterial endosymbionts are particularly useful when their metabolites complement nutrient-poor diets of hosts. Examples are *Buchnera* (γ-Proteobacteria) providing aphids with essential amino acids (*Baumann et al., 1995*; *Shigenobu et al., 2000*), *Wolbachia* (α-Proteobacteria) producing vitamin B for *Cimex lectularius* bedbugs (*Hosokawa et al., 2010*; *Nikoh et al., 2014*), *Baumannia* and *Sulcia* (γ-Proteobacteria) providing sharpshooters (*Homalodisca coagulata*) with vitamins and amino acids (*Wu et al., 2006*), *Nardonella* (γ-Proteobacteria) providing beetles with tyrosine required for cuticle formation (*Anbutsu et al., 2017*), and *Stammera* bacteria allowing leaf beetles to decompose pectin (*Salem et al., 2017*). A recent comparative analysis confirmed that nutrient supplementation often drives evolution towards host dependence especially when symbionts are vertically transmitted (*Fisher et al., 2017*).

The social insects with superorganismal colonies, characterized by permanent physiologically and morphologically differentiated castes, appear particularly amenable for hosting specialized bacterial symbionts. Previous studies have documented amino acid provisioning by *Blochmannia* (γ-Proteobacteria) hosted by *Camponotus* carpenter ants (*Feldhaar et al., 2007*), and suggested that bacterial symbionts provision *Cephalotes* turtle ants with essential amino acids (*Hu et al., 2018*) and *Cardiocondyla* ants with useful intermediate metabolites (*Klein et al., 2016*). Honeybees were further suggested to rely on several specialized gut bacteria for carbohydrate breakdown of ingested pollen and nectar (*Engel et al., 2012*) and fungus-growing termites were discovered to have caste-specific microbiomes depending on whether individuals ingest plant material (mainly decaying wood) or only farmed fungus (*Poulsen et al., 2014*). Finally, both bees and termites rely on gut microbes to provide them with acetate that can cover up to 100% of their metabolic needs (*Odelson and Breznak, 1983*; *Zheng et al., 2017*).

The leaf-cutting ants are the crown group of the attine fungus-growing ants, a monophyletic tribe that evolved 55–60 MYA when their ancestor switched from a hunter-gatherer lifestyle to an exclusive fungal diet (*Nygaard et al., 2016*; *Branstetter et al., 2017*). The evolutionarily derived attine lineages rear fully domesticated and co-adapted fungal cultivars that provide the ant farmers with

specialized hyphal tips (gongylidia) containing mostly carbohydrates and lipids that the workers harvest and digest (*De Fine Licht et al., 2014*; *Quinlan and Cherrett, 1979*). The ant brood is completely dependent on the ingestion of fungal biomass (*Hölldobler and Wilson, 1990*), but workers may ingest and assimilate liquids as well (*Littledyke and Cherrett, 1976*; *Shik et al., 2018*). However, similar to other ants, they cannot ingest solid plant or animal fragments that they collect to provision their fungus gardens because a sieve in the infrabuccal cavity filters out any particles in excess of ca. 100 µm (*Mueller et al., 2001*). This obligate reciprocity between cultivation and nutrition facilitated further innovations in the terminal clade of *Acromyrmex* and *Atta* leaf-cutting ants, which evolved 15-20 MYA (*Nygaard et al., 2016*; *Branstetter et al., 2017*). These two genera obtained functionally polyploid cultivars (*Kooij et al., 2015*), adopted multiple queen-mating so their colonies became genetic chimeras (*Villesen et al., 2002*), and became herbivores with massive ecological footprints in Latin America (*Schultz and Brady, 2008*; *Mehdiabadi and Schultz, 2010*; *Schiøtt et al., 2010*; *Leal et al., 2014*; *Shik et al., 2014*).

Previous studies have shown that *Acromyrmex* and *Atta* leaf-cutting ants harbor low-diversity microbiomes, which include *Wolbachia* (only in *Acromyrmex*), Mollicutes and hindgut *Rhizobiales* (*Van Borm et al., 2002*; *Andersen et al., 2012*; *Sapountzis et al., 2015*; *Meirelles et al., 2016*), symbionts that were inferred to possibly complement the nitrogen-poor diets of *Acromyrmex* leaf-cutting ants (*Sapountzis et al., 2015*). Depending on the actual species studied, Mollicutes – tiny bacteria that lack a cell-wall – can often be found as abundant endosymbionts in up to 100% of leaf-cutting ant colonies (*Sapountzis et al., 2015*; *Meirelles et al., 2016*; *Zhukova et al., 2017*), but the absence of in-depth genomic studies has precluded more than speculation about their putative roles as either parasites (*Meirelles et al., 2016*) or mutualists (*Sapountzis et al., 2015*).

To clarify the functional metabolic properties of attine-associated Mollicutes, we mapped the abundances of the two most common strains, *EntAcro1* and *EntAcro10* (cf. *Sapountzis et al., 2015*), in thirteen Panamanian fungus-growing ant species and compared these abundances with the typical spectrum of forage-material that different fungus-farming ants collect and use as compost to manure their fungus-gardens (*Kooij et al., 2014a*; *Leal and Oliveira, 2000*; *Shik et al., 2016*). The Panamanian fauna of attine ants encompasses nine of the 17 known genera, including the three most basal genera (*Apterostigma*, *Mycocepurus* and *Myrmicocrypta*), two other basal genera (*Cyphomyrmex* and *Mycetophylax*) being more closely related to the *Trachymyrmex* and *Sericomyrmex* lineages that arose and diversified while rearing gongylidia-bearing cultivars, and finally the *Atta* and *Acromyrmex* leaf-cutting ants who came to practice fungus-farming at an 'industrial' scale (*Branstetter et al., 2017*; *Schultz and Brady, 2008*; *Mueller et al., 1998*). To explore nutritional mechanisms underlying putative mutualistic functions of these bacteria, and their association with changes in scale of farming over evolutionary time, we isolated and sequenced the *EntAcro1* and *EntAcro10* symbionts (*Sapountzis et al., 2015*; *Zhukova et al., 2017*). We subsequently compared their draft genomes with ten published genomes of Mollicutes associated with insect hosts having specialized diets, as well as with several other Mollicutes genome-sequences to assess enrichments and losses of gene categories and metabolic pathways.

For *EntAcro1*, where the genomic data suggested the most advanced mutualistic functions, we measured expression levels of bacterial transporter genes related to the decomposition of plant-derived compounds and ant genes related to the uptake of exogenous acetate, an end-product of Mollicutes' anaerobic metabolism. Our genome comparisons also allowed us to evaluate arginine decomposition functions and defense mechanisms against bacteriophage attack, assuming that: i) variable food-borne arginine supplementation by the fungal cultivar (*Nygaard et al., 2016*; *Nygaard et al., 2011*) may have offered a niche to both Mollicutes symbionts to convergently evolve similar mutualistic interactions with attine ants, and ii) the abundant and specific bacteriophage sequences that we obtained in the libraries of the *EntAcro1* symbiont indicate that these bacteria have been under selection to maintain costly defenses because extracellular life in the gut lumen likely exposes them to frequent phage encounters.

## Results

### Metagenome sequencing and phylogenomics

De-novo assembly, annotation and phylogenetic binning of contigs generated from *Ac. echinatior* fecal fluid and *Ap. dentigerum* fat body produced a single bin of contigs for each symbiont, confirming them to be valid bacterial species that we will henceforth refer to as *EntAcro1A* and *EntAcro10A*; *Supplementary file 1A*). The predicted coding sequences gave top matches with previously sequenced Mollicutes strains (*Supplementary file 1B*) and each of the bins had a single rRNA operon organized as 16S, 5S, 23S (*Figure 1—figure supplement 1*), similar to closely related *Spiroplasma* (*Ku et al., 2013*; *Lo et al., 2013*; *Chang et al., 2014*) and identical to OTUs in a previous 16S phylogeny (*Sapountzis et al., 2015*). Additional bins (*B*; see *Supplementary file 1B and 1C*) did not contain relevant bacterial sequences and were not considered further, similar to the *EntAcro1C* bin that contained bacteriophage sequences with similarity to members of the Gokushovirinae subfamily (*Microviridae* family), a phage lineage known to infect *Spiroplasma* (*Chipman et al., 1998*; *Supplementary file 1*). Further analyses of the annotated coding sequences confirmed that *EntAcro1A* and *EntAcro10A* represented discrete draft genomes of *EntAcro1* and *EntAcro10* with no or very few missing genes. These genomes had 758 and 776 coding sequences, respectively, and genome sizes of less than 0.9 Kb based on the annotation features (*Supplementary file 1A*; *Figure 1—figure supplement 1*).

A total of 59 published Mollicutes genomes were used for phylogenomic reconstructions after their predicted proteins gave clear matches to the *EntAcro1* and *EntAcro10* amino acid sequences (*Supplementary file 1B*). This produced nearly identical trees after maximum likelihood (*Figure 1*) and Bayesian analysis of nucleotide and amino acid sequences (*Supplementary file 2*; *Figure 1—figure supplement 2*) and revealed that *EntAcro1* and *EntAcro10* belong to the Entomoplasmatales group, confirming earlier 16S assignments (*Sapountzis et al., 2015*). This clade contains *Spiroplasma* and *Mesoplasma* bacteria associated with insects and plants and *Mycoplasma* bacteria known to be mammalian pathogens. Sister-group relationships showed that *EntAcro1* and *EntAcro10* are not closely related and thus likely to have been independently acquired as attine ant symbionts. *EntAcro10* is a relatively basal *Spiroplasma*-like species, a genus with pathogenic, mutualistic and yet unknown interactions with mostly arthropod hosts. However, *EntAcro1* is sister to the *Mesoplasma/ Mycoplasma* clade and relatively similar to one of the very few Entomoplasmatales known to be associated with plant hosts in a clade that otherwise consists of vertebrate pathogens. The more distant sister clades are also predominantly pathogenic.

### Substrate utilization and reconstruction of metabolic pathways

We restricted our comparative evaluations and hypotheses testing to the *Spiroplasma* and *Mesoplasma* symbionts associated with insect hosts (the attine symbionts highlighted in dark yellow in *Figure 1* and the ones in between) and plants (*M. florum*). Functional annotations (eggNOG database) showed that strains clustered primarily based on metabolic genes and secondarily according to shared identity for informational genes (transcription, translation and recombination/repair processes), which both correlated with host associations (*Figure 1—figure supplement 3*; *Figure 1— figure supplement 3—source data 1*). Functional gene-similarities were confirmed by Mantel tests showing that the Euclidean dissimilarity matrix of orthogroups was more strongly associated with phylogenomic distances between insect hosts ($r^2$ = 0.298, p=0.036) than with phylogenomic distances between bacterial species (*Figure 1*; $r^2$ = 0.201, p=0.133), suggesting that many genes that adapt Entomoplasmatales symbionts to their hosts have been horizontally acquired. Metabolic reconstructions (KEGG) further suggested that all Entomoplasmatales are facultative anaerobes, because they lack the genes encoding TCA cycle enzymes and are thus universally incapable of oxidative phosphorylation. However, *EntAcro1* likely lost its aerobic abilities completely because pyruvate dehydrogenase genes are also missing. Differences in metabolism between the two attine symbionts and other insect-associated *Spiroplasma/Mesoplasma* strains were primarily found in pathways mediating catabolism of glycerol, dihydroxyacetone (DHA), citrate, arginine, and N-Acetyl-glucosamine (GlcNAc) (*Figure 2* and *Supplementary file 3*).

Metabolic pathway reconstructions were consistent with *EntAcro10* being a less specifically adapted symbiont than *EntAcro1* (*Figure 2*). Inferences of this kind, based on direct similarity

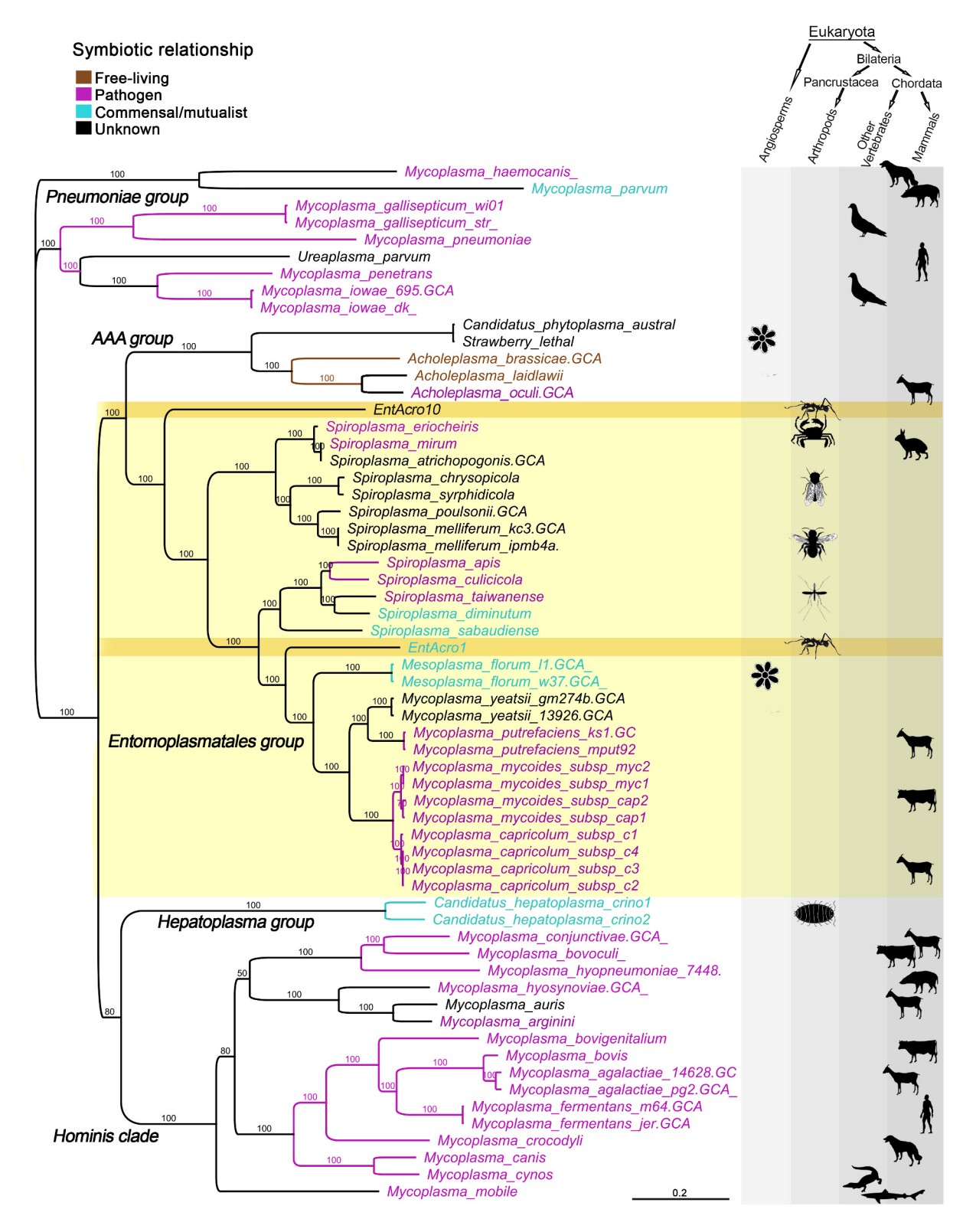

**Figure 1.** Unrooted phylogeny illustrating the evolutionary relationships and host associations of 59 strains of the bacterial class Mollicutes for which sequenced genomes were available. The Maximum likelihood tree was constructed using a concatenated amino acid alignment of 65 single-copy orthologs (*Supplementary file 2*) giving 100% bootstrap support for almost all branches. Color-coded Latin names specify different types of symbiotic associations (top left, *Supplementary file 4*) and host associations are illustrated by images in the right-hand columns. The light yellow background

*Figure 1 continued on next page*

*Figure 1 continued*

highlights all representatives of the Entomoplasmatales and the two ant symbionts *EntAcro1* and *EntAcro10* are marked with a dark yellow background. Black bold-faced text towards the left refers to subclades of Mollicutes identified in a previous phylogenomics study (*Leclercq et al., 2014*).

DOI: https://doi.org/10.7554/eLife.39209.003

The following source data and figure supplements are available for figure 1:

**Figure supplement 1.** Overview of the assembled draft genome contigs of *EntAcro1A* (top) and *EntAcro10A* (bottom).

DOI: https://doi.org/10.7554/eLife.39209.004

**Figure supplement 2.** Alternative unrooted phylogenies based on the concatenated nucleotide and protein alignments of the 65 single-copy orthologs showing the phylogenetic relationships among 59 Mollicutes strains with sequenced genomes.

DOI: https://doi.org/10.7554/eLife.39209.005

**Figure supplement 3.** Principal Components Analyses (PCA) visualizing gene content similarities of *EntAcro1*, *EntAcro10* and the eleven most closely related Mollicutes genomes.

DOI: https://doi.org/10.7554/eLife.39209.006

**Figure supplement 3—source data 1.** Functional annotation results for the 14 closely related insect-associated Mollicutes genomes using the bacterial eggnog (bactNOG) database.

DOI: https://doi.org/10.7554/eLife.39209.007

between bacterial genomic databases, may not be fully accurate because gene-families encoding metabolic transporters evolve rather rapidly so the actual transported substrates may no longer be identical. However, the draft genomes that we obtained were sufficiently complete to provide reasonable confidence for reconstruction of operational metabolic transporters through the plasma membrane, the associated metabolic pathways inside the bacterial cells, and the metabolic end-/by-products involved (*Figure 2*). We found that *EntAcro10* can utilize glycerol from ant host cells and monosaccharides which are likely derived from fungus-garden metabolites or juices (glucose/fructose) ingested during foraging (*Figure 2*). We also found an arginine transporter and metabolic genes indicating that *EntAcro10* can decompose arginine. This finding is of interest because the attine ants lost the ability to synthesize this nitrogen-rich amino acid (highest nitrogen to carbon ratio of all amino acids) when fungus-farming evolved, so they obtain arginine from the fungal cultivar and potentially also from ingested fruit juice and plant sap in the herbivorous crown-group leaf-cutting ants (*Kooij et al., 2014a*; *Winter et al., 2015*).

The evolutionarily derived *EntAcro1* symbiont had all the transporters and pathways identified for *EntAcro10* but also novel ones that would appear to be particularly adaptive for an abundant extracellular endosymbiont in the gut lumen (*Sapountzis et al., 2015*) of leaf-cutting ants. First, it has a GlcNAc transporter likely importing chitin monomers, a relatively common metabolic pathway in Mollicutes found in 85 of the 177 genomes that we considered. The possession of a chitin importer is relevant because chitin is one of the most abundant compounds in the ants' fungal diet (*Figure 2*), particularly in the leaf-cutting ants whose cultivars have modified and likely thicker cell walls than the cultivars of phylogenetically more basal attine ants (*Nygaard et al., 2016*). Second, *EntAcro1* has a rather unique citrate transporter indicative of a rare catabolic pathway observed in only four of the 177 available Mollicutes genomes; *Figure 2—figure supplement 1*). Further examination of the five or six citrate utilization genes involved in this pathway (*citS, citC, citD, citE, citF* and potentially *citG*; see *Figure 2—figure supplement 1*) showed that all these genes are extremely rare across the Mollicutes genomes and present similarities to genes from bacterial classes outside the Mollicutes (e.g. Firmicutes and Clostridia) suggesting they were horizontally obtained (*Figure 2—figure supplement 1*). Anaerobic citrate fermentation, or co-fermentation of glucose/citrate, using the *citS-citF* operon will produce acetate (*Pudlik and Lolkema, 2011*; *Starrenburg and Hugenholtz, 1991*), which is likely imported by eukaryote cells to fuel metabolism (*Figure 2—figure supplement 2*) or stored in the fat body cells (*Figure 2—figure supplement 3*). The citrate pathway thus appears to reflect that leaf-cutting ants can utilize the citrate metabolite that they are known to ingest in substantial quantities as plant sap when cutting fresh leaves (*Littledyke and Cherrett, 1976*) and in the form of other juices when drinking from freshly fallen fruit (*De Fine Licht and Boomsma, 2010*; *Evison and Ratnieks, 2007*).

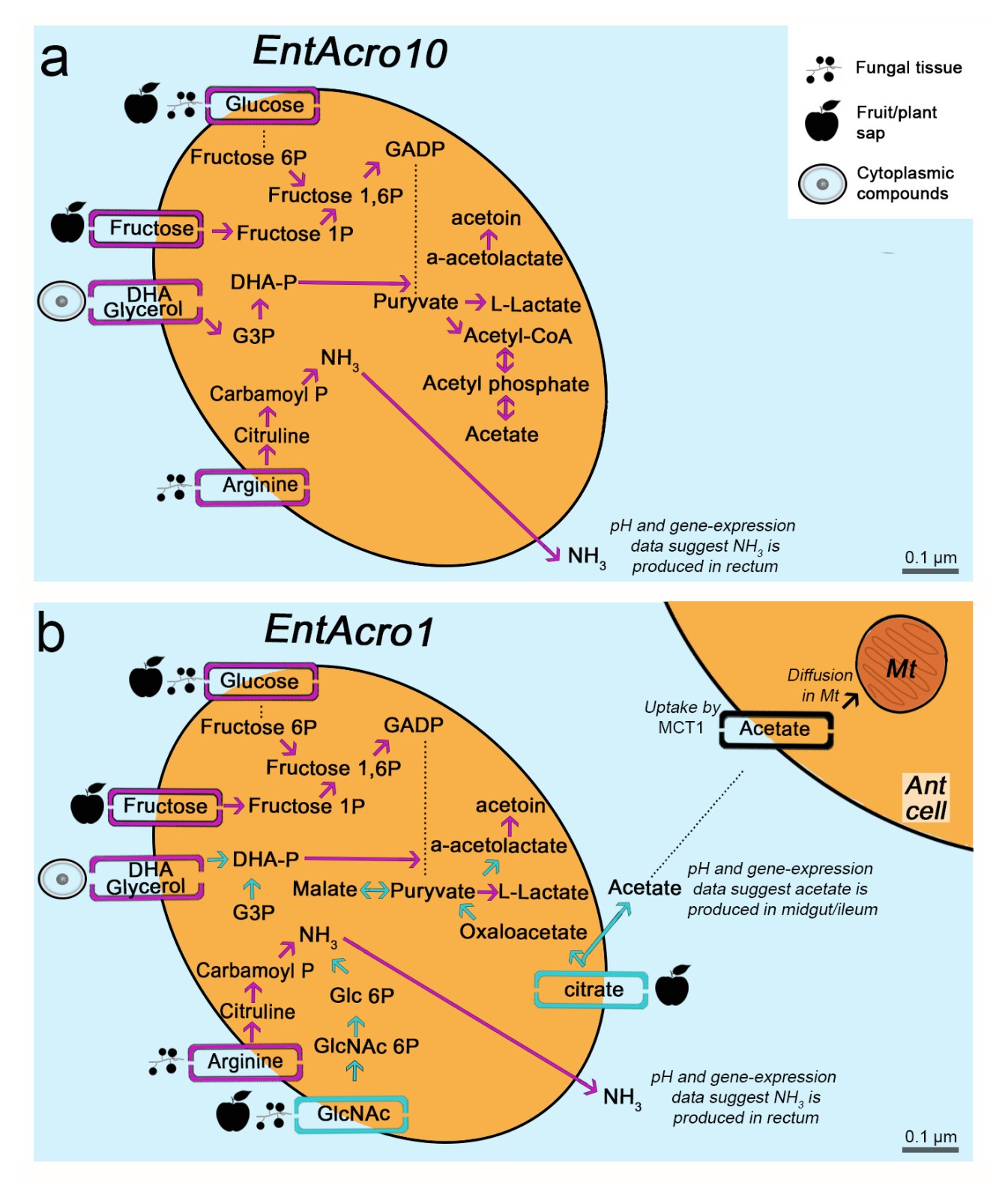

**Figure 2.** Morphological and metabolic characteristics of the *EntAcro10* and *EntAcro1* symbionts of attine fungus-growing ants. (a) Schematic *EntAcro10* cell with pink arrows representing KEGG pathway reconstructions based on intact genes coding for proteins metabolizing specific compounds and pink rectangles crossing the plasma membrane representing predicted transporter genes that import metabolites from the ant gut lumen where fungal biomass and imbibed fluids are digested (cf symbols next to the transporter rectangles). GlcNAc is present in fungal tissues

*Figure 2 continued on next page*

Figure 2 continued

(*Pérez and Ribas, 2013*) or plant material that may have been decomposed by the fungal cultivar, citrate and glucose/fructose are present in fruit juice and possibly plant sap that ants drink (*Liu et al., 2012*; *Medlicott and Thompson, 1985*), arginine is mainly provided by the ingested fungal tissues or juices from leaves and fruits (*Nygaard et al., 2016*; *De Fine Licht et al., 2014*; *Nygaard et al., 2011*; *Winter et al., 2015*), and glycerol is commonly available in the cytoplasm of eukaryote cells (*Grubmüller et al., 2014*; *Joseph et al., 2008*; *Monniot et al., 2012*; *Sun et al., 2003*). Dotted lines between metabolic products represent intermediate genes and other metabolic products not shown. Arginine metabolism produces $NH_3$, which will be excreted and help the bacteria survive the acidic conditions (*Figure 4—figure supplement 1*) to end up in the fecal droplets used to manure the ants fungus-garden. The bacterial cell shape and the estimated size (scale bars) were obtained using TEM data from a previous study (*Sapountzis et al., 2015*). (b) Schematic representation of the *EntAcro1* cell with the functional KEGG pathway reconstructions and predicted transporters inferred from the *EntAcro1* genome that are identical to *EntAcro10* in panel 'a' as pink arrows. Unique additional pathways and transporters only found in *EntAcro1* are drawn in green, and include the anaerobic citrate fermentation pathway, which produces acetate that can be taken up by ant cells, and the GlcNAc pathway that results in byproducts that can enter glycolysis in the bacterial cells.

DOI: https://doi.org/10.7554/eLife.39209.008

The following figure supplements are available for figure 2:

**Figure supplement 1.** Citrate utilization genes in Mollicutes.

DOI: https://doi.org/10.7554/eLife.39209.009

**Figure supplement 2.** Synopsis of the fate of extracellular acetate (on the left) and citrate (on the right) when they enter a eukaryotic cell, based on the available literature.

DOI: https://doi.org/10.7554/eLife.39209.010

**Figure supplement 3.** Schematic representation of two ant fat body cells and the process of storing acetate in the lipids of the fat body cells.

DOI: https://doi.org/10.7554/eLife.39209.011

## Resource acquisition, gene expression and inferred Mollicutes functions

Screening *EntAcro* abundances in worker bodies across the Panamanian attine ants showed distinct patterns of prevalence. *EntAcro10* was present in most attine species investigated such that there were no significant differences across the entire set of 13 species (planned contrasts, z = −0.62, p=1.00). However, *EntAcro1* was almost exclusively found in the leaf-cutting ants (planned contrasts, z = 2.88, p=0.016) with their closest Panamanian relative *T. cornetzi* and yeast-growing *C. rimosus* as the only (partial) exceptions. At the same time *EntAcro10* abundances in leaf-cutting ants were lower than *EntAcro1* abundances albeit only marginally so (planned contrasts, z = −2.56, p=0.045) (*Figure 3A*; *Figure 3—source data 1*). This pattern implies that the appearance of *EntAcro1* is correlated with changes in the spectrum of substrates that the farming ants provide to their fungus gardens (*Figure 3B*; *Figure 3—source data 2*), with fresh leaf, fruit and flower provisioning dominating in the leaf-cutting ants and the phylogenetically more basal attines collecting primarily detritus-based substrates such as insect frass and wood chips (*Figure 3—figure supplement 1*). These differences were variably (non)significant per forage category (*Figure 3—figure supplement 1*), but bacterial abundances and foraging preferences generally covaried for *EntAcro1* (Mantel test; $r^2$ = 0.447, p=0.012) but not for *EntAcro10* (Mantel test; $r^2$ = −0.127, p=0.882). These results suggest that the derived metabolic pathways of *EntAcro1* (*Figure 2*) may have been associated with the expansion of the scale of fungus farming and the adoption of functional herbivory in the leaf-cutting ants.

The abundances of *EntAcro1* bacteria are known to be highly variable, between lab and field colonies, between replicate colonies that seem otherwise fully comparable, and between nestmate workers of both *Atta* and *Acromyrmex* leaf-cutting ants (*Sapountzis et al., 2015*; *Zhukova et al., 2017*). To evaluate the potential mutualistic benefit of acetate production by *EntAcro1* (*Figure 2—figure supplement 2*), we experimentally manipulated *EntAcro1* abundances in lab colonies of *Ac. echinatior* and measured rates of acetate uptake in the same ants. We compared ants maintained on their normal fungus garden diet with ants on a sugar/citrate replacement diet with or without antibiotics, known from pilot experiments to remove most Mollicutes from the guts and associated organ systems. We quantified *EntAcro1* abundances by measuring the number of transcripts of the bacterial housekeeping gene *ftsZ* and the rates of acetate uptake by ant host cells by measuring the expression of *MCT1* (*Figure 4*; *Figure 4—source data 1*), a gene encoding a plasma membrane protein that imports acetate in eukaryotic cells; *Figure 2—figure supplement 2*). We found that these variables were positively correlated (ρ = 0.57, p<0.001), suggesting that acetate production by *EntAcro1* boosts acetate uptake by the ants who likely convert acetate directly into ATP (*Figure 2—figure supplement 2*). This inference is somewhat tentative because tetracycline can impair

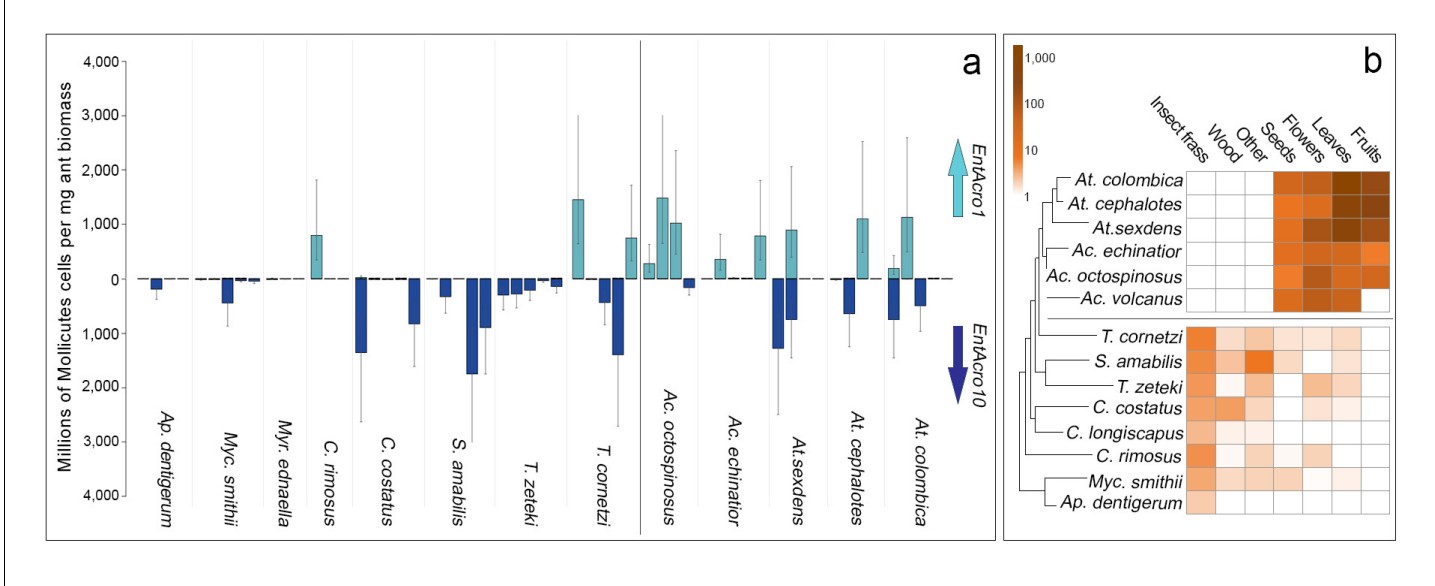

**Figure 3.** Absolute abundances of *EntAcro1* and *EntAcro10* and foraging substrate preferences across thirteen species of Panamanian fungus-growing ants spanning the entire attine ant phylogenetic tree. (a) Millions of Mollicutes cells per mg of ant worker biomass (means ± 95% CI based on two technical replicates) across colonies for *EntAcro1* (light blue upward arrow) and *EntAcro10* (dark blue downward arrow). The vertical line between *Ac. octospinosus* and *T. cornetzi* separates the leaf-cutting ants (right) from the non-leafcutting ants (left). (b) Foraging substrate preferences among Panamanian attine ant species presented as mean frequency heat-maps of substrate categories collected per hour for five leaf-cutting ant species (*Kooij et al., 2014a*) and seven non-leaf-cutting ant species (this study). The tree has been modified from (*Branstetter et al., 2017*; *Schultz and Brady, 2008*) and the horizontal line separates the leaf-cutting ants (top) which forage mostly on fresh plant material, from the more basal attine ants (bottom) which forage mostly on detritus based material (*Figure 3—figure supplement 1*).

DOI: https://doi.org/10.7554/eLife.39209.012

The following source data and figure supplement are available for figure 3:

**Source data 1.** QPCR data for the absolute quantification of EntAcro strains in fungus-growing ants.

DOI: https://doi.org/10.7554/eLife.39209.014

**Source data 2.** Merged ant foraging data used for our analyses of data originating from *Sapountzis et al. (2015)* and the present study.

DOI: https://doi.org/10.7554/eLife.39209.015

**Figure supplement 1.** Principal Components Analysis (PCA) capturing the covariances between attine ant species (colored dots) and foraging substrate preferences (colored arrows) across the Panamanian attine ant hosts of *EntAcro1* and *EntAcro10* endosymbionts (data from *Figure 3B*, *Figure 3— source data 2*).

DOI: https://doi.org/10.7554/eLife.39209.013

mitochondrial function (*Moullan et al., 2015*) and thus overall metabolic functionality, a confounder that could have been measured by tracking a specific mitochondrial protein. The fact that the antibiotics data point in *Figure 4* directly extended the trend obtained from the treatments without antibiotics (sugar/citrate diet) and the control (fungal diet) suggests that this confounding effect has been minor but further work will be needed to validate this result. Conversion of citrate to acetate by the *EntAcro1* symbiont would be consistent with the general observation that leaf-cutting ants sustain much higher levels of worker activity, both inside nests and while foraging, than phylogenetically more basal attine ants (*Kooij et al., 2014a*).

To better understand the function of the arginine processing genes (*Figure 2*), we returned to our comparative genomic data for the attine symbionts *EntAcro1* and *EntAcro10* and the ten *Spiroplasma* species associated with other insects (*Figure 5*). The observation (*Figure 4—figure supplement 1*; *Figure 4—source data 1*) that the arginine transporter of *EntAcro1* was most highly expressed in the hindgut lumen suggests that *EntAcro1* cells may need to decompose arginine in exceedingly low pH conditions (≤4). This physiological tolerance may represent a fine-tuned mutualistic service, as became clear when we evaluated patterns of gene expression of all four transporters mediating catabolism of key resources in the gut system and associated ant tissues. The predicted

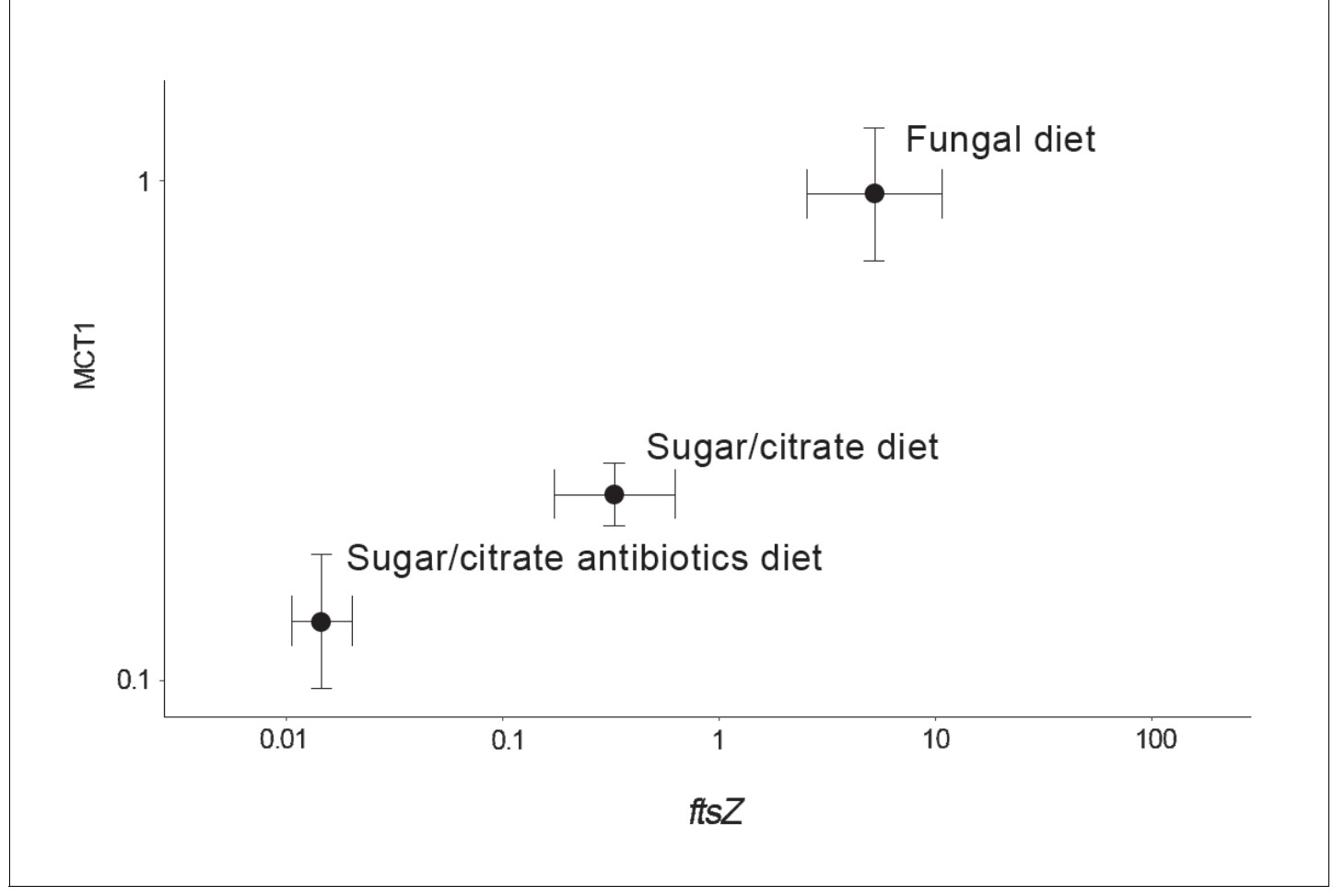

**Figure 4.** Correlation between the abundance of the *EntAcro1* symbiont (expression of the *EntAcro1* housekeeping gene *ftsZ*) and the activity of the predicted plasma-membrane Monocarboxylate Transporter-1 gene (*MCT1*) in the midgut and fat body tissues of *Ac. echinatior* workers. Both axes are logarithmic and express numbers of transcripts normalized relative to the expression of the housekeeping ant gene *rpl7*). Control ants were provided with their natural fungal diet, while manipulated nestmates were kept for seven days without a fungus garden but with access to a glucose/citrate solution with or without antibiotics. Values are means (±SEs) of 20 pooled workers replicated twice (technical replicates) across four colonies. The correlation between the two transcript abundances was significant, with the first component of a PCA analysis explaining 91% of the variation. A non-parametric Spearman correlation analysis produced a similarly significant result ($\rho$ = 0.57, p<0.001), but correlation tests between *ftsZ* transcripts and the expression of the other MCT-like genes present in the ant genome did not produce significant correlations (*Supplementary file 6*).
DOI: https://doi.org/10.7554/eLife.39209.016

The following source data and figure supplements are available for figure 4:

**Source data 1.** Expression data for the monocarboxylate transporter genes examined in our study.
DOI: https://doi.org/10.7554/eLife.39209.019

**Figure supplement 1.** Differential expression and importance of the four substrate utilization transporter genes of *EntAcro1* that we analyzed.
DOI: https://doi.org/10.7554/eLife.39209.017

**Figure supplement 1—source data 1.** Expression data for the bacterial transporter genes examined in our study.
DOI: https://doi.org/10.7554/eLife.39209.018

transporters of arginine, citrate, GlcNAc and glycerol or DHA (*Figure 4—figure supplement 1*; *Figure 4—source data 1*) were expressed throughout the guts and associated organs of *Ac. echinatior* workers, but their expression levels differed across abdominal tissues possibly in response to a steep gradient from pH seven in the midgut and fat body cells where the glycerol transporter is highly expressed, via pH five in the ileum and pH four in the rectum (*Erthal et al., 2004*) where the arginine transporter is highly expressed (*Figure 4—figure supplement 1*; *Figure 4—source data 1*). Earlier work has indicated that utilization of the *citS-citF* operon is most efficient at or just above pH 5.5

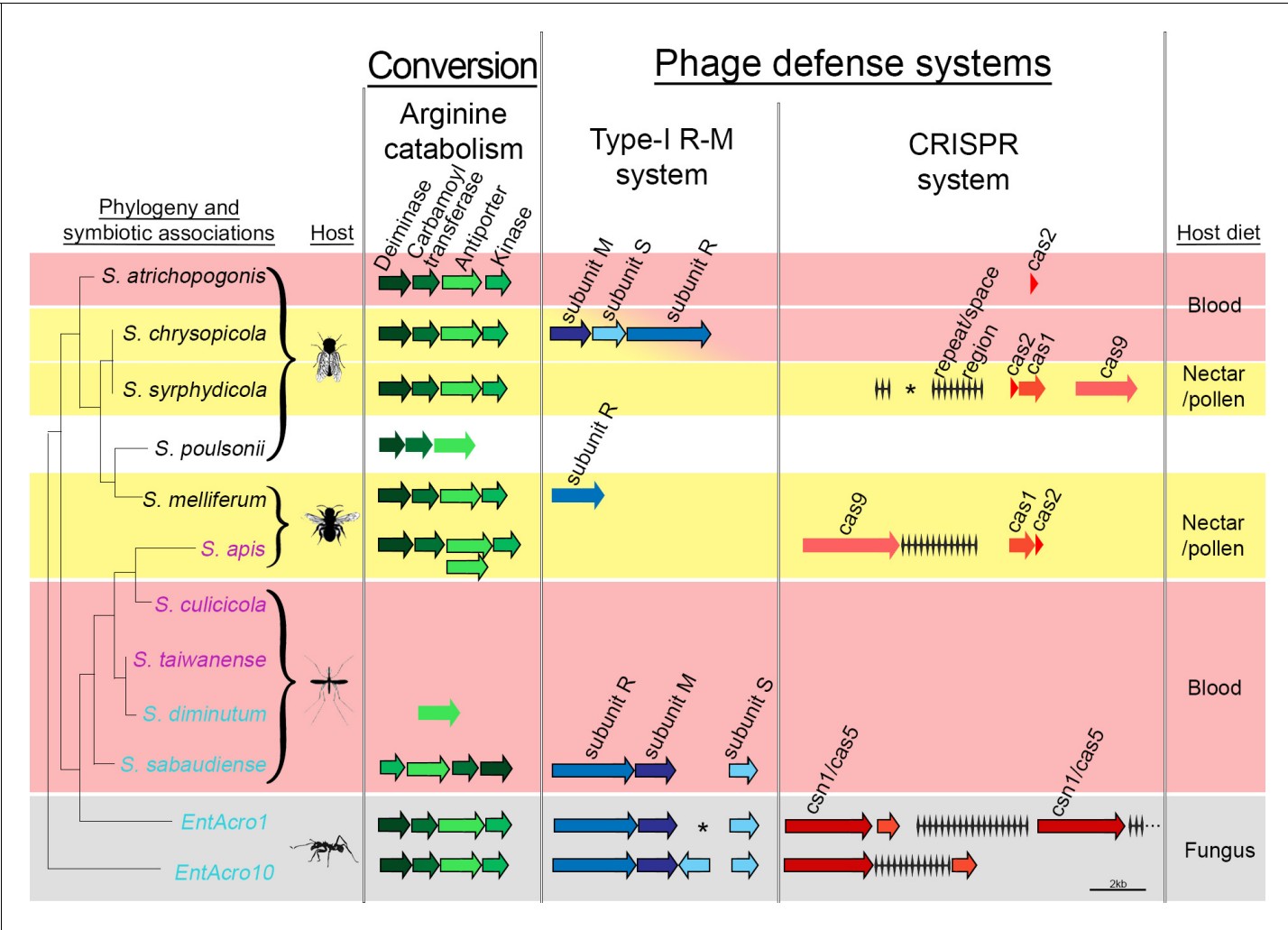

**Figure 5.** Organization of the arginine catabolism operon and two groups of bacteriophage-defense related genes (Type-I R-M and CRISPR systems) in *EntAcro1* and *EntAcro10* and closely related *Spiroplasma* symbionts of other insect hosts. The tree on the left and the color-coded species names specify different types of association (as in ***Figure 1*** with pink representing confirmed pathogenicity, light blue representing confirmed non-pathogenicity, and black indicating unknown but likely non-pathogenic; ***Supplementary file 4***). Background colors highlight specialized host diets: fungus (gray), blood (red), nectar/pollen (yellow; mixed in deerflies hosting *S. chrysopicola* where males feed on nectar/pollen and females on blood) and unclear (white; food of *Drosophila* host is unknown but unlikely to be a specialized diet). Genes in each pathway have different shades of a single color and sizes of arrows and spaces between them are proportional to actual gene sizes relative to the 2 kb scale bar (bottom right). Gene orientations correspond to the direction of arrowheads and all genes are presented with their common names as identified by our RAST annotations (***Supplementary file 3***). Gene-arrows have a black outline when all core genes for a focal pathway are present, indicating it is functionally active. The asterisks for the Type-I R-M system in *EntAcro1* and the CRISPR system in the symbiont of *S. syrphydicola* indicate that the gap between the two arrows is not proportional to the scale bar but much wider.

DOI: https://doi.org/10.7554/eLife.39209.020

The following figure supplements are available for figure 5:

**Figure supplement 1.** Phylogeny of all amino acid transporters in *Saccharomyces*, *Agaricus bicolor*, *Coprinopsis cinerea*, *Schizophyllum commune* and the *Leucogaricus* symbionts of *Atta*, *Acromyrmex* and *C. costatus*.
DOI: https://doi.org/10.7554/eLife.39209.021

**Figure supplement 2.** KEGG metabolic reconstructions based on the intact genes present in the *Acromyrmex*, *Solenopsis*, *Apis mellifera* and *Anopheles gambiae* genomes, together constituting the urea cycle.
DOI: https://doi.org/10.7554/eLife.39209.022

**Figure supplement 3.** Overview of the potential metabolic complementarity, inferred by our genomic reconstruction, between fungus-growing ants, their fungal cultivar, and their *EntAcro* symbionts.
DOI: https://doi.org/10.7554/eLife.39209.023

(*Magni et al., 1999*; *Sánchez et al., 2008*), suggesting that citrate catabolism by *EntAcro1* cells happens primarily in the midgut and in ileum where the pH is optimal for that function (*Figure 4—figure supplement 1*), which leaves arginine decomposition as the main terminal digestion process in the hindgut where pH is low.

Decomposition of all available hindgut arginine into $NH_3$ just before the anaerobic *EntAcro1* symbionts would die from exposure to aerobic conditions via the ants' fecal fluid would ensure that manure of the fungus-garden provides nitrogen in its most readily available form for fungal protein synthesis. The conversion of excess arginine to ammonia in the ant hindgut may thus resolve a potential mutualistic mismatch because the attine fungal cultivars have generic amino acid transporters, but they lack specialized arginine transporters to process environmental arginine, similar to other basidiomycete fungi (*Figure 5—figure supplement 1*). In general, ammonia is the preferred nitrogen source for fungal growth (*Ahmad et al., 1990*; *Abril and Bucher, 2004*), so any increase in the ammonia to arginine ratio of fecal fluid manure would benefit the farming symbiosis as a whole. At the same time this conversion prevents nitrogen waste, as would happen when excess arginine were to be deposited on a fungal cultivar primarily adapted to using simpler nitrogen sources.

## Bacteriophage defense genes

We found clear evidence for both *EntAcro1* and *EntAcro10* having two intact bacterial defense systems to ward off phage attack, a Type-1 R-M system and a CRISPR pathway (*Figure 5*). Genes belonging to both defense systems are often horizontally transmitted among bacteria (*Labrie et al., 2010*) and maintaining them is costly (*Stern et al., 2010*; *Vasu and Nagaraja, 2013*; *Vale et al., 2015*; *Burstein et al., 2016*), so these defenses are primarily expected in bacterial species that face consistent threats of phage attack without being severely resource-constrained. Gene-level comparisons with the other *Spiroplasma* symbionts showed that none of them had the same dual defense system against phage attack. Type I R-M system genes were present only in non-pathogenic Mollicutes strains, similar to arginine catabolism pathways being restricted to *Spiroplasma* strains with non-pathogenic host associations, except for the reputedly pathogenic *S. apis* (*Figure 5*; *Supplementary file 4*). The generally rarer CRISPR system was complete (i.e. both CRISPR repeat/space regions and cas genes being present) only in *S. chrysopicola*, *S.apis*, and the two ant associated *EntAcro* strains.

Our finding that the two attine ant symbionts are unusually well protected is consistent with them being vulnerable to phage attack when they reach high densities in the gut lumen. We did indeed find *Spiroplasma*-specific phages of the Gokushovirinae in the contig bin 'C' of the *EntAcro1* symbiont (*Supplementary file 1C*) isolated from fecal fluid. Not finding these phage sequences in *EntAcro10* might reflect that these bacteria were isolated from the fat body of *A. dentigerum* where they are intracellular symbionts and that titers of *EntAcro10* were very low (*Figure 3*). These functional inferences are tentative, but potentially of significant interest, so we will return to them below.

## Discussion

At the Panama site where we conducted our study, the *EntAcro1* and *EntAcro10* symbionts are the most common Entomoplasmatales strains associated with attine ants and they represent the majority of sequence reads (>40% jointly for both *EntAcro* symbionts that were obtained from these ants in field colonies and >50% in captive colonies fed *ad libitum*; *Sapountzis et al., 2015*). Our study thus captured much of the qualitative and quantitative biodiversity of abdominal Mollicutes endosymbionts. We show that these two symbionts are phylogenetically distant and therefore evolved independently (*Figure 1*), but that their gene contents reflect convergent adjustment to life as ant symbionts when compared to related *Mesoplasma* and *Spiroplasma* bacteria associated with other arthropods or plants (*Figure 1—figure supplement 3*). These convergences primarily relate to carbohydrate metabolism, consistent with patterns of bacterial adaptation being generally based on substrate utilization (*Lo et al., 2015*; *Pál et al., 2005*).

Several of the differences in metabolic genes between *EntAcro1* and *EntAcro10* relative to the *Mesoplasma* and *Spiroplasma* symbionts of the remaining insects emanate from rare genes that catabolize substrates such as citrate and arginine. These are complex molecules that attine ants and their fungal cultivars are likely to provide to their bacterial endosymbionts, but that are unlikely to be present in the food of other insects. None of the leaf-cutting ant colonies with a high prevalence

of Mollicutes symbionts ever showed any signs of pathogenicity, even when ants had several million Mollicutes cells in their bodies. Earlier studies left this issue open (*Sapountzis et al., 2015*; *Meirelles et al., 2016*), but the combined results of our present study clearly suggest that *EntAcro1* and *EntAcro10* are co-adapted mutualists and that their metabolic pathways shed novel light on several poorly understood aspects of the highly complex attine fungus-farming symbiosis.

### The arginine recycling niche of Spiroplasma-like abdominal symbionts

The loss of the arginine synthesis pathway in the basal attine ants (*Nygaard et al., 2011*; *Suen et al., 2011*) has been instrumental in making their fungus-farming symbiosis obligate (*Nygaard et al., 2016*). The selection regime that caused this loss remains unknown (*Nygaard et al., 2016*; *Ješovnik et al., 2016*), but it is reasonable to assume that outsourcing the production of this most nitrogen-rich amino acid to fungal cultivars gave complementary efficiency benefits even though it also generated symbiotic dependency. For symbiotic division of labor to be sustainable under variable environmental conditions, average levels of fungal arginine production would have to be higher than the minimally sufficient level to avoid occasional windows of fatal shortage in the symbiosis as a whole. Symbiotic dependency may thus have created a niche for Mollicutes symbionts to ensure that surplus arginine is recycled as $NH_3$ to provide the most efficient manure for new garden growth.

Tissues of adult insects no longer grow and may thus only need small amounts of nitrogen for maintenance, so passing on excess $NH_3$ to fungus gardens via fecal fluid would have unambiguous benefits for the complex mutualism as a whole (*Schiøtt et al., 2010*). This conjecture was recently confirmed for *Atta* workers in an independent study showing that workers fed with ammonium nitrate (the protonated form of ammonia) transfer nitrogen via their fecal fluid to the fungus garden (*Shik et al., 2018*). Mollicutes-assisted garden manuring would thus imply that any surplus nitrogen remains a stable resource for new fungal protein synthesis and thus growth of the ant brood that only ingests fungal food. This underlines that driving-agency in obligate farming mutualisms is ambiguous. A shorter explanation of the prudency of this co-adaptation is that garden fungi domesticated ants to maintain and disperse them, and that they benefitted from the ants domesticating Mollicutes to ensure not a single nitrogen atom is wasted and their keepers could (later) utilize external resources such as citrate to work harder to enhance fungal growth.

The only other ant lineage in which Entomoplasmatales (Mollicutes) endosymbionts have so far been abundantly found are the army ants (*Funaro et al., 2011*). These ants are exclusive predators of mostly invertebrate prey (*Kronauer, 2009*) and 16S rDNA sequences of their Mollicutes symbionts suggested they are closely related to *EntAcro1* but rather distantly to *EntAcro10* (*Funaro et al., 2011*). It is intriguing that the Dacetine sister lineage of the fungus-farming ants are also specialized predators (*Branstetter et al., 2017*; *Ward et al., 2015*). It would thus be interesting to clarify whether also Dacetine ants have Entomoplasmatales symbionts, how (un)related they would be to the *EntAcro* symbionts of the fungus-growing ants, and whether army ants acquired their Mollicutes horizontally from preying upon on attine ants (*Powell and Clark, 2004*).

Broader consideration of the general feeding ecology of all insects hosting Mollicutes symbionts revealed that they are mostly specialized on nutritionally deficient but protein-rich diets, such as vertebrate blood (flies and mosquitoes) and pollen/nectar (bees and flies). The pollen/nectar feeders all had *Spiroplasma* symbionts with complete arginine catabolism pathways, suggesting they might similarly convert excess dietary arginine into $NH_3$, although the absence of functional studies precludes speculation about the type of mutualistic advantage yielded by this conversion (*de Groot, 1952*; *Vrzal et al., 2010*; *Uchida, 1993*; *Honeybee Genome Sequencing Consortium, 2006*; *Nene et al., 2007*; *Figure 5—figure supplement 2*). Thus, while excreted $NH_3$ would appear to have a clear mutualistic function for manuring attine fungus-gardens (*Figure 5—figure supplement 3*), the benefits of $NH_3$ production in the gut system of bees and some mosquitoes is less clear. Overall, it seems that primarily non-pathogenic *Spiroplasma* strains may have been selected to catabolize host-food-associated arginine (*Figure 5*; *Supplementary file 3*; *Supplementary file 4*), but this provisional inference needs explicit functional verification by artificial diet experiments and selective removal of symbionts to quantify putative changes in arginine and ammonia titers.

### *Did the acquisition of* EntAcro1 *facilitate the emergence of large-scale herbivory?*

Our results indicate that *EntAcro1* was acquired as additional symbiont to *EntAcro10* relatively shortly before the leaf-cutting ants evolved and that *EntAcro1* supplements already available arginine recycling with novel pathways allowing ant workers to process non-fungal metabolites (*Figure 2*; *Figure 4*; *Figure 5—figure supplement 3*). Because citrate catabolism genes do not exist in *EntAcro10* the lower attine ants may be generally unable to convert plant-derived citrate or glucose/citrate into acetate. The acquisition of *EntAcro1* thus likely allowed the ant farmers to tap into additional non-fungal resources to maintain higher metabolic rates. These differences match what is generally known about the increases in farming scale, foraging activity, and garden growth-rates when moving from the basal attine ants to the derived branches of the attine phylogenetic tree (*Kooij et al., 2015*; *Shik et al., 2014*; *Shik et al., 2016*). It is interesting that one of the closest relatives of *EntAcro1* (*M. florum*) is associated with plants (*Figure 1*), which could suggest that this symbiont was acquired when the ancestors of the leaf-cutting ants started to forage on live plant material. However, confirmation of this hypothesis would require the closest relatives of *EntAcro1* to be associated with American Angiosperms. The few current records are from citrus trees of Asian origin (*Liu et al., 2012*), so substantial sampling effort will be needed to investigate this possible plant association.

The timing of the acquisition of *EntAcro1* is intriguing. It was recently shown (*Branstetter et al., 2017*) that a monophyletic crown group of the attine ants evolved in Central/North America following colonization of this subcontinent by a single South-American attine ancestor 22–27 MYA, well before the isthmus of Panama closed. The timing of isthmus closure is controversial with some maintaining that it happened as recently as ca. 3 MYA (*O'Dea et al., 2016*), while other studies indicate it may have been as early as the mid-Miocene ca. 13–15 MYA (*Bacon et al., 2015*; *Montes et al., 2015*). A recent study on army ants, whose queens are wingless and thus dependent on solid landbridges for dispersal, indicated colonization of Central-North America 4–7 MYA (*Winston et al., 2017*). Also this dating is much later than the inferred timing of the first attine ant arrival in what was then the Central-North American subcontinent (*Branstetter et al., 2017*).

The implication is that most of the higher attine ant radiation happened on a novel subcontinent that was devoid of attine ants. In this context it is interesting that we also found *EntAcro1* in some field colonies of *T. cornetzi* (*Figure 3*), a higher non-leaf-cutting ant representing the most basal attine branch that colonized Central-North America (*Branstetter et al., 2017*). This suggests that *EntAcro1* may have been domesticated in Central-North America in response to the founding lineage encountering new ecological opportunities perhaps including plants with *EntAcro1*-like symbionts. Also here, larger sampling efforts will be needed to verify whether endosymbiotic microbiome signatures of this major biogeographic vicariance event continue to be found across the extant attine ants of Central/North and South America. If *EntAcro1* was domesticated just before or during the transition to active herbivory it may have somehow facilitated the transition to industrial-scale farming as presently found in the *Atta* and *Acromyrmex* leaf-cutting ants throughout the Americas.

### The costs and benefits of defending domesticated bacterial symbionts in the gut

We found that both *EntAcro1* and *EntAcro10* have dual, fully intact cellular defenses against phage attack, consisting of a R-M (Restriction Modification) Type one system and a CRISPR pathway (*Figure 5*). We obtained a substantial number of phage sequences specific for *Spiroplasma*-like bacteria (the *EntAcro1C* bin; *Supplementary file 1C*) and our data show that the abundances of particularly *EntAcro1* inside ant bodies can be very high (*Zhukova et al., 2017*) (*Figure 3*). The mere presence of phage defensive mechanisms is not surprising because high clonal bacterial densities make bacteriophage attack efficient and rewarding. However, what makes the intactness of two complete pathways of the two *EntAcro* symbionts interesting is that none of the related *Spiroplasmas* associated with other insects has two such intact pathways. Both types of phage defenses are last resort systems, operating only when a phage has already broken through the bacterial cell membrane and has released its DNA into the bacterial cell. Recent work has clarified that these bacterial defenses are analogous to a non-specific innate immune system (R-M) and an adaptive (trainable) immune system

for recognizing specific viral DNA (CRISPR) (*Vasu and Nagaraja, 2013*; *Seed, 2015*), and that the two systems can operate synergistically (*Dupuis et al., 2013*). While ca. 90% of all bacterial genomes have at least one Restriction Modification system (*Vasu and Nagaraja, 2013*), less than 44% of bacterial genomes appear to have a CRISPR system (*Burstein et al., 2016*; *Makarova et al., 2011*) and these specific defenses are typically absent in obligate symbionts (*Burstein et al., 2016*).

It is increasingly documented that both types of phage-defense systems are likely to have fitness costs (*Stern et al., 2010*; *Vasu and Nagaraja, 2013*; *Vale et al., 2015*; *Burstein et al., 2016*). These costs may be expressed as slower growth in the absence of phages, somewhat analogous to the costs of autoimmune errors (*Stern et al., 2010*) and would be consistent with many bacterial lineages losing CRISPR genes relatively easily (*Burstein et al., 2016*). Lack of exposure explains why intracellular symbionts rarely have phage defenses compared to gut-lumen symbionts with much higher exposure to phages, which would also explain the presence of these systems in *EntAcro1*, an abundant gut lumen symbiont (*Sapountzis et al., 2015*). However the presence of these systems may also depend on a general trade-off between maintenance and growth. Preservation by active defense is much more likely to be a naturally selected priority for a vertically transmitted mutualist than for a pathogen selected to infect other colonies at the highest possible rate. Although tissue localization and proximate mechanisms such as the need to maintain chromosomal stability and recombination are important in determining the likelihood of acquisition and loss of phage-defense genes (*Vasu and Nagaraja, 2013*), the ultimate evolutionary cost-benefit argument is compelling enough to be spelled out for explicit testing in the future.

The endosymbiont-host interactions that we document include several feedback loops that should allow the ants to regulate *EntAcro* symbiont densities upwards or downwards depending on the overall costs and benefits of their services, similar to other hosts such as aphids which are able to control their intracellular *Buchnera* symbionts (*Wilkinson et al., 2007*; *Russell et al., 2014*). This underlines that phenotypic mechanisms for using symbiont services based on immediate cost-benefit ratios apply for both intra- and extra-cellular symbionts. Providing *EntAcro1* and *EntAcro10* symbionts with sufficient resources to maintain a full complement of phage-defense systems when they reach high densities would then appear to be a cost-efficient strategy to secure mutualistic services. This is because the only available route for propagation to future generations of a Mollicutes strain is to help maximize the colony's production of dispersing virgin queens (*Meirelles et al., 2016*). This would be achieved by host-induced optimization of bacterial titers rather than by maximal rates of bacterial cell division, in contrast to commensal or pathogenic bacteria that remain under selection to primarily maximize their rates of horizontal transmission (*Frank, 1996*). Comparative experimental tests measuring the phage-attack-sensitivity of intracellular and extracellular symbionts with and without phage defenses could be a way to verify these expectations that are consistent with the results presented here and with general evolutionary theory on levels-of-selection, efficiency of transmission, and the expression of competitive symbiont traits (e.g. *Frank, 2012*).

## Note added in proof

A study by *Gupta et al. (2018)* that came online while our article was in the final stage of proof-checking performed a phylogenomic analysis of 121 conserved protein sequences in 32 Mollicutes genomes, of which some overlapped with the Mollicutes compared in our analyses. The Gupta et al. study complements the results presented in our paper by: 1. Confirming that *EntAcro1* and *EntAcro10* have very different origins, 2. Confirming that *EntAcro10* is the most basal branch of the Entomoplasmacaeae and Spiroplasmataceae consistent with our *Figure 1*, 3. Confirming that the closest relative of *EntAcro1* is associated with plants – in their study a *Mesoplasma lactucae* isolated from lettuce corroborating the suggestion that the ancestors of the leafcutter ants may have acquired *EntAcro1* from plants on which they foraged, 4. Showing that citrus-associated *Mesoplasma florum* belongs to a more derived lineage closer to *Mycoplasma* strains consistent with our *Figure 1*, and 5. Placing the *Mesoplasma lactucae* strain in a new genus *Edwardiiplasma* together with *EntAcro1*.

## Materials and methods

### Rearing and handling of ant colonies

*Apterostigma dentigerum* and *Acromyrmex echinatior* colonies were collected in Gamboa, Panama and maintained in rearing rooms at 25 °C and 70% relative humidity during a 12:12 hr photoperiod at the Centre for Social Evolution, University of Copenhagen, Denmark. The *Acromyrmex* colony (Ae331) used in this study had been kept under laboratory conditions for 8 years (collected in 2007), while the *Apterostigma* field colonies were all collected in May 2015, brought to the lab and sampled within the first two weeks from their field collection date. Some aspects of the composition of attine ant microbiota associated with the intestinal system may change after they are reared in the lab for a number of years, but the *EntAcro1* and *EntAcro10* symbionts are little affected and remain the two dominant Mollicutes symbionts across the attine ants at our field site in Gamboa, Panama also after colonies are transferred to the lab (*Sapountzis et al., 2015*; *Zhukova et al., 2017*).

### Bacterial isolation, genome amplification and DNA extraction

Colony Ae331 from which we isolated the *EntAcro1* symbiont had been screened previously by 16S-Miseq sequencing and targeted 16S-PCR reactions, which showed its workers had high titers of *EntAcro1* and no detectable traces of the possible alternative strains *EntAcro2* or *EntAcro10* (*Sapountzis et al., 2015*). To obtain a pure sample of *EntAcro10* bacteria we used the lower attine ant *Ap. dentigerum* and performed an initial survey on 20 freshly collected colonies by extracting DNA from whole workers and performing PCR with *EntAcro10* specific primers (*Sapountzis et al., 2015*). This showed that workers from one colony (RMMA150520-03) were carrying the *EntAcro10* strain without any other Mollicutes being detectable. Prior to the further isolation of the two symbiont strains, a series of *Acromyrmex* and *Apterostigma* workers from these two colonies were anesthetized and surface sterilized by submergence in 70% ethanol for 1 min, after which they were rinsed twice in autoclaved MilliQ water, submerged in 50% bleach for 2 min, and rinsed again twice in autoclaved MilliQ water.

For the bacterial isolations we used a previously described protocol with some modifications (*Ellegaard et al., 2013*; *Iturbe-Ormaetxe et al., 2011*). For *EntAcro1* we obtained ca. 50 fecal droplets from *Ac. echinatior* workers under a laminar flow hood (*Kooij et al., 2014b*) and deposited them in sterile petri dishes using sterile forceps, after which they were jointly suspended in 1000 μL cold SPG Buffer (218 mM sucrose, 3.8 mM $KH_2PO_4$, 7.2 mM $K_2HPO_4$, 4.9 mM l-glutamate, pH 7.2) and transferred to 1.5 ml Eppendorf tubes. To isolate *EntAcro10*, we dissected fat body cells from ca. 25 surface sterilized *Ap. dentigerum* workers under a stereomicroscope and immediately transferred them to a sterile 15 mL glass homogenizer (Wheaton) on ice, along with 1000 μL of cold SPG buffer. Using a glass pestle, we disrupted the tissues on ice and immediately transferred them into a new 1.5 ml Eppendorf tube.

The samples from both ant species were centrifuged at 4 °C for 15 min at 3,200 g after which the supernatant was transferred to new 1.5 ml microcentrifuge tubes and centrifuged again with the same settings. The supernatant was subsequently purified through a 5 μm (Acrodisc) and a 2.7 μm (Whatman) syringe filter, and finally through a 1.3 μm (Acrodisc) filter before transfer to a new 1.5 ml tube followed by centrifugation for 20 min at 18,000 g at 4 °C. The supernatant was discarded and the pellets (bacterial cells) were re-eluted in 5 μl SPG buffer. Approximately, 1 μl was then used for Multiple Displacement Amplification (MDA) to obtain whole genomic DNA using the Qiagen REPLI-g Midi Kit following the manufacturer's instructions. A blank reaction using sterile water as template instead of 1 μl of bacterial cell suspension was included in the same protocol to check for bacterial contaminations with eubacterial 515F/806R primers after the entire procedure was completed, which showed no detectable 16S amplicons.

The purified bacterial pellets used for MDA and subsequent dilution of the amplified DNA were subjected to PCR using the 16S generic primers 515F and 806R (*Caporaso et al., 2012*) as previously described (*Bourtzis and Miller, 2006*), purified using the Invitek kit (Westberg, Germany), and sent to MWG (Germany) for Sanger sequencing. After we had confirmed that the 16S amplicons were of Mollicutes' origin and that the chromatographs showed no signs of other bacterial 16S rDNA sequences, DNA was further purified using the Qiagen mini spin kit following the manufacturer's instructions. The extracted DNA was then quantified for both the *Ac. echinatior* and *Ap.*

*dentigerum* sample using a Nanodrop spectrophotometer and sent to seqIT (Germany) where libraries were generated from 100 to 200 ng of DNA using the Nextera XT kit (Illumina, USA). Finally, MiSeq sequencing was performed at 2 times 250 bp read length, which generated approximately 3,000,000 reads per sample.

## Assembly, annotation and quality controls

The Nextera adaptors used for the library construction were removed from the fastq files using Trim Galore (Babraham Institute) and the filtered reads were checked with FastQC (*Andrews, 2016*). We then used the SPAdes Genome Assembler (version 3.5.0) to generate a de novo assembly using the '–careful' option which reduced the number of mismatches and short indels before running MismatchCorrector with kmer sizes of 21, 33, 55 and 77 to obtain a consensus assembly based on four individual assemblies (*Bankevich et al., 2012*). We then used the Burrows-Wheeler Aligner (BWA) to map reads to the assembled contigs (*Li and Durbin, 2009*), which produced a SAM file that was further analyzed using SAMTOOLS and converted to a BAM file that could be analyzed with Bamviewer v1.2.11 (*Carver et al., 2010*). The assembled contigs were further checked for errors using the Reapr v1.0.18 software (*Hunt et al., 2013*) and contigs that had less than 9x coverage or were smaller than 250 bp were removed. The final set of assembled contigs *Supplementary file 1*) was deposited in the NCBI Genome submission portal under accession numbers SAMN06251630 and SAMN06251631.

Genes for each contig were predicted using the RAST annotation server (*Aziz et al., 2008*) after which predicted amino acid sequences were compared to a local database Uniref100 using BLASTP v2.2.28+ (evalue <1e-15, percentage identities > 30%), and top matches with the assembled contigs were phylogenetically binned (*Supplementary file 1*). This grouped the contigs belonging to Mollicutes together in what we refer to as 'A' bins allowing further evaluation of the strain-specific RAST annotations (*Simão et al., 2015*). We functionally annotated the protein sequences using a stand-alone version of InterproScan v-5.17–56.0 with SUPERFAMILY, Pfam, ProSiteProfiles, Coils, ProSitePatterns, TIGRFAM, Hamap and ProDom (*Jones et al., 2014*) and Phobius (*Käll et al., 2007*) (http://www.cbs.dtu.dk/services/SignalP/ and http://phobius.sbc.su.se/) to predict signal peptide and transmembrane domains. To identify and compare metabolic pathways we used the KAAS tool (*Moriya et al., 2007*) provided by the KEGG database (*Kanehisa and Goto, 2000*; *Kanehisa et al., 2010*) together with the BLAST algorithm and the single best hit (SBH) procedure with default settings.

## Phylogenomic analyses

For the phylogenomic reconstructions, we first downloaded all 176 available Mollicutes genomes that were present in the Ensembl database (page accessed in January 2016; (*Yates, 2016*)) and created a merged customized BLAST database, which we used to compare the predicted amino acid proteins of *EntAcro1A* and *EntAcro10A*. We included only complete genomes and thus excluded the partially sequenced *Entomoplasma melaleucae* genome. The BLAST comparisons (*Altschul et al., 1990*) using an e-value of 1e-15 as cutoff and a percentage identity of 30%, revealed clear similarities between our two *EntAcro* symbionts and 59 previously genome-sequenced Mollicutes strains. We therefore used their genome sequences to define the orthologous single-copy protein-coding genes using the Orthofinder software (*Emms and Kelly, 2015*) which resulted in 73 genes being available for phylogenomic analyses. Both the nucleotide and amino acid sequences of these genes were extracted for each of the two *EntAcro* symbionts, which allowed the construction of gene-specific alignments using MUSCLE v3.8.31 (*Edgar, 2004*). These alignments were subsequently refined using the trimAl software, which removed all positions with gaps in 10% or more of the sequences unless this left fewer than 50% of the original sequences (*Capella-Gutiérrez et al., 2009*). The filtered alignments were further tested for recombination using the Phipack software (*Bruen et al., 2006*) and for nucleotide saturation using the Xie test implemented in DAMBE5 (*Xia and Xie, 2001*). For the 65 genes that remained (*Supplementary file 2*) individual alignments were concatenated using Amas 0.98 (*Borowiec, 2016*) and the appropriate substitution models for the nucleotide and protein alignments were selected after testing them with jmodeltest v2.1.7 and prottest v3.4 (*Darriba et al., 2011*).

We used the nucleotide and amino acid alignments for the two *EntAcro* symbionts and the other 59 Mollicutes genomes to reconstruct phylogenomic trees with maximum likelihood (ML) and Bayesian methods. For the ML analyses we used the RaxML software v.8.2.10 (*Stamatakis et al., 2008*) with specified partitions of the concatenated alignments, which produced 65 partitions (one for each gene) using the GTR model with invariable sites for the nucleotide alignment and the LG + IG model for the amino acid alignments (*Le and Gascuel, 2008*) after 1000 bootstrap sampling replications. For the Bayesian inferences, we used the MrBayes v3.2.6 software (*Ronquist et al., 2012*) applying the same nucleotide and amino acid models as above. The concatenated alignments were specified for each gene-specific partition and a variable rate of sequence evolution (ratepr) was allowed for each of them. Initially, five chains were run for one million generations and statistical samples were taken every 1000 generations, after which the analyses were repeated to cover a total of 10 million generations, because analysis of the effective sample sizes showed under-sampling in the initial trials. This produced an appropriate deviation of split frequencies (a standard measure in mrbayes which allows examination of how similar the calculated trees of two independent runs were) for the nucleotide (0.0014) and protein alignments (0.00001), with values well below the 0.01 threshold recommended as evidence for sufficient convergence and effective sample sizes exceeding 100 in all cases. All trees were further processed using FigTree v1.4.2 implemented in Geneious R7.1.1 (*Kearse et al., 2012*). The clipart images used in *Figure 1* were either obtained from wikimedia commons (https://commons.wikimedia.org/), or phylopic (http://phylopic.org/), available under a Public Domain License or drawn in Adobe Photoshop CS6.

## Comparative genomics

We used the bactNOG database v4.5 (page accessed April 2016) to find clusters of orthologous genes and to compare the predicted proteins with HMMER v3.1.b1 (*Eddy, 2011*; *Huerta-Cepas et al., 2016*). To obtain specific comparisons among Mollicutes genomes based on different numbers of genes assigned to distinct functional categories via bactNOG, we obtained ratio estimates for the number of genes in each functional category by counting the number of genes assigned to specific orthogroups and dividing by the total number of annotated genes (proportional abundances), and used these data as input for Principal Component Analysis (PCA) using the 'stats' package in RStudio (v. 1.0.136).

We used Mantel tests to compare phylogenomic distances based on orthologous genes for insect hosts or bacterial symbionts with the overall genome dissimilarities. We focused on the eleven Mollicutes genomes (and their insect hosts) that we used for most of our genomic comparisons, because they were closely related. These were *EntAcro1*, *EntAcro10*, *S. sabaudiense*, *S. diminutum*, *S. culicicola*, *S. taiwanense*, *S. apis*, *S. melliferum*, *S. atrichopogonis*, *S. chrysopicola* and *S. syrphidicola*. We originally also included the *S. poulsonii* genome but the extremely high number of transposases (identified as ENOG410907Q, ENOG4105YCW, ENOG4105Y09, ENOG4105DQ6; *Paredes et al., 2015*) made this genome a clear outlier (also in the PCA ordinations presented in *Figure 1—figure supplement 3*) so we excluded it from further comparative analyses. We thus set out to compare 11 *Spiroplasma* and *EntAcro* strains with respect to: 1) their gene functional categories, 2) their bacterial phylogeny, and 3) their hosts' phylogeny. To compare gene functional annotations of the Mollicutes genomes, we used their orthogroup proportional abundances (see above) and created a dissimilarity matrix using Euclidean distance in R. For the bacterial phylogeny matrix, we used the phylogenomic distances constructed for this study based on the amino acid sequences of 65 orthologs (see above and *Figure 1*). For the genome-based host phylogeny matrix, we used a recent publication that constructed a global phylogeny for the holometabolous insects (*Song et al., 2016*). Whenever possible we used the same host species that the Mollicutes bacteria are associated with (*Apis mellifera*, *Aedes albopictus*, *Culex pipiens*), but when that was not possible we used the closest respresentatives within the same taxonomic clade for which a sequenced genome was available: *Simosyrphus grandicornis* instead of *Eristalis arbustorum*, *Cydistomyia duplonata* instead of *Chrysops. sp.*, *Culicoides arakawae* instead of *Atrichopogon*, and *Solenopisis geminata* instead of *Ac. echinatior* and *Ap.dentigerum* (*Ward et al., 2015*). A host phylogeny for our matrix comparisons was then reconstructed using the selected mitochondrial genome sequences retrieved from NCBI (Accession numbers available in Supplementary Material of (*Song et al., 2016*) using MUSCLE v3.8.31 (*Edgar, 2004*) and FastTree v1.0 (*Price et al., 2009*). The final Mantel tests were performed in R using 10,000 permutations, which produced correlations between: 1. the functional annotation

matrix and the bacterial phylogeny matrix, 2. the functional annotation matrix and the host phylogeny matrix, and 3. the host and the bacterial phylogeny matrices.

## Artificial diet experiments and reverse transcription quantitative PCR

To verify the putative mutualistic functions of the *EntAcro1* symbiont, we pursued two experimental approaches. First, we experimentally reduced the abundance of the symbionts in the bodies of ant workers and measured whether this had a negative effect on the acetate uptake activity of ant cells. We achieved this reduction by keeping ants on artificial diets that we knew from pilot experiments were either marginal or outright discouraging for the maintenance of Mollicutes endosymbionts. For these experiments we used four different colonies of *A. echinatior*: Ae150, Ae331, Ae360 and Ae507, which had been kept in the lab for 15, 9, 8 and 5 years respectively, before the experiment. Ant workers were removed from their fungus gardens and placed in sterile petri dishes with an inverted screw-lid in the middle filled with the nutrition-medium of choice. Using the known composition of mango fruit juice (*Medlicott and Thompson, 1985*), a resource that leaf-cutting ants at our field site regularly utilize, we created an artificial diet consisting of 140 mM sucrose, 130.6 mM fructose, 70 mM glucose and 4 mM sodium citrate (Sigma aldrich, Denmark). This diet was offered to the ants for seven days either in pure form or with 1 mg/ml tetracycline and 1 mg/ml rifampicin added. As controls we used workers picked directly from their fungus-garden.

We examined the expression of relevant genes in three type of tissues of ant workers: (1) approximately 50 fecal droplets from the rectum; (2) fat body, midgut and part of the ileum tissues of 20 ants; and (3) heads and thoraxes of 5 ants to represent the remaining body parts as controls. We excluded all samples originating from the heads and thoraxes from further analyses, because we never detected any bacterial gene expression in them except for a single sample (*ftsZ* gene expression in colony Ae360). We validated the effect of our diet manipulation by measuring the expression of *ftsZ*, a single copy bacterial gene that we amplified with qRT-PCR using *EntAcro1*-specific primers (*Supplementary file 5*). To evaluate acetate import activity by ant cells, we measured the expression of an MCT-1 ortholog that has been demonstrated to mediate acetate uptake in animals (*Kirat and Kato, 2006*; *Moschen et al., 2012*) as well as nine other MCT-like genes in the *Ac. echinatior* genome that are predicted to import multiple short-chain fatty acids (SCFAs) potentially including acetate (www.uniprot.org; *Supplementary file 6*). To measure the expression of MCT-like genes and the number of *EntAcro1* cells (by measuring expression of the bacterial housekeeping gene *ftsZ*), we only used fat body and midgut tissue samples (which also included part of the ileum), since we could not detect any expression of MCT-like genes in the rectum lumen (not surprising because the rectum lumen has no ant cells). To evaluate the statistical significance of the correlation between expression of each MCT or MCT-like gene and *ftsZ*, we used non-parametric Spearman correlation tests on the deltadelta $C_T$ values.

In a second set of experiments using the same type of ant tissues as above, we compared the expression of four predicted *EntAcro1* transporter genes (arginine, GlcNAc, citrate, glycerol or DHA; see *Figure 4—figure supplement 1*) in the midgut and fat body tissues relative to the (hindgut) rectum lumen to obtain insight in the metabolic activity of *EntAcro1* symbionts throughout the alimentary tract where pH is known to change from ca. neutral to acid (ca. pH 4) conditions just before defecation (*Erthal et al., 2004*). For this experiment, we also sampled 50 ant workers from the same four *Ac. echinatior* colonies (Ae150, Ae331, Ae360 and Ae507). To test the effect of the gene identity and tissue compartment on the expression data, we used an ANOVA linear model with the deltadeltaCt values (ratio of target gene expression/*ftsZ* gene expression) from the qPCR as response variable and the predicted transporter gene identity (arginine, citrate, GlcNAc and glycerol/DHA), the tissue identity (midgut/fat body or hindgut) and their interaction as fixed factors using the function 'aov' in R (*Chambers and Hastie, 1992*). We evaluated significant differences across groups (transporter gene and gut compartment) using post-hoc comparisons as planned contrasts and Bonferroni corrections based on the 'glht' function in the 'multcomp' package (*Hothorn et al., 2008*). Before performing these tests, we visually examined normality of the data distribution ('hist' command) and confirmed impressions of no significant deviations with Shapiro–Wilk tests, which gave no indications of heteroskedasticity or deviations from normality that would compromise the validity of parametric statistics.

Technical procedures in both experiments were as follows: All tissues were dissected and collected in ice-cold RNAlater and stored at −80°C until extraction after which total RNA was extracted

using an RNeasy minikit (Qiagen, Germany). Fifty-five ng of extracted RNA was treated with RQ1 RNase-free DNase I (Promega Corporation, Madison, WI), and 50 ng of the resulting product was reverse transcribed with an iScript RT kit (Bio-Rad Hercules, CA, USA) to obtain first-strand cDNA. As a negative control, the remainder of the DNase-treated RNA was examined by PCR under the same conditions. All gene-specific PCRs were performed on cDNA, DNase-treated RNA, ant DNA, and water as in previous procedures (*Andersen et al., 2012*; *Sapountzis et al., 2015*). For the qPCR reactions we used the SYBR Premix Ex Taq PCR mix (TaKaRa Bio Inc., St. Germain en Laye, France) on the Mx3000P system (Stratagene, Santa Clara, CA, USA). Reactions took place in a final volume of 20 µl containing 10 µl buffer, 8.3 µl sterile double-distilled water (ddH$_2$O), 0.4 µl of each primer (10 mM), 0.4 µl ROX standard, and 0.5 µl template cDNA. PCR conditions were as follows: denaturation for 2 min at 94°C, followed by 40 cycles of 30 s at 94°C, 30 s at the annealing temperature (see *Supplementary file 5* for primers used), and 30 s at 72°C, followed by dissociation curve analysis. All quantitative PCRs (qPCRs) were replicated and each run included two negative controls with no added template.

To generate the delta values for the qPCR analyses, we used the fold change method, based on a standard curve with PCR products in a tenfold dilution series of known concentrations to calculate the PCR efficiency of each primer pair using the REST software (*Pfaffl, 2002*). Data were then imported into R and expressed as deltadelta $C_T$ values, that is, as fold changes relative to the *Ac. echinatior* specific *rpl7* housekeeping gene for the ant host and the *ftsZ* gene specific for the *EntAcro1* symbiont (*Pfaffl, 2001*).

### Estimation of absolute abundances of EntAcro1 *and* EntAcro10 *in ant hosts and other statistical analyses*

Pooled abdominal tissues from five ant workers were collected from five lab colonies of *At. colombica*, four colonies of *At. cephalotes*, four colonies of *At. sexdens*, five colonies of *Ac. echinatior*, four colonies of *Ac. octospinosus*, five colonies of *T. cornetzi*, five colonies of *T. zeteki*, five colonies of *S. amabilis* and four colonies of *Ap. dentigerum*. These samples were supplemented with ca 5–15 whole worker gaster samples (abdomens minus the first segments that are integrated in the thorax or form the well-known hymenopteran constriction that has no organs) of attine ant species that were too small to dissect: five lab colonies of *C. costatus*, three colonies of *C. rimosus*, three colonies of *Myr. ednaella* and five colonies of *Myc. smithii*. All samples were stored in −20°C and DNA was extracted based on previously described methods (*Sapountzis et al., 2015*).

We estimated the abundances of Mollicutes symbiont DNA using qPCR with primers targeting the 16S gene of *EntAcro1* (*Sapountzis et al., 2015*) and *EntAcro10* (*Supplementary file 5*). For each 16S gene that we analyzed, the initial template concentration was calculated from a standard curve with PCR product in tenfold dilution series of known concentration, as quantified by nanodrop. Since our genomic data showed that one 16S copy corresponded to one *EntAcro1* or one *EntAcro10* cell, we first calculated the results as numbers of bacterial symbiont cells per ant. We chose not to normalize the data using a specific single copy ant gene (such as EF-1a used in other studies; *Sapountzis et al., 2015*), because samples were from different tissues, that is, either dissected fat bodies and midguts for the ant species with larger body size or whole gasters for the small-bodied species. However, we did obtain data on species-specific fresh-weight body mass of workers and used them to approximately scale the bacterial cell counts. To estimate mean worker body mass per attine ant species, we weighed five random workers from three colonies from each species to obtain the following average values: *A. colombica* 4 mg, *A. cephalotes* 4.1 mg, *A. sexdens* 4.7 mg, *A. echinatior* 9.7 mg, *A. octospinosus* 12.5 mg, *T. cornetzi* 1 mg, *T. zeteki* 1.9 mg, *S. amabilis* 1.2 mg, *C. costatus* 0.1 mg, *C. rimosus* 0.4 mg, *M. ednaella* 0.3 mg, *M. smithii* 0.2 mg and *A. dentigerum* 2.2 mg.

To compare the abundance levels of *EntAcro1* and *EntAcro10* per unit of worker body mass, we used a generalized (negative binomial) linear model (GLM) with the function 'glm.nb' in the package 'MASS' (*Venables and Ripley, 2002*). This model was a better fit than a GLM model with gamma or Poisson distribution when we compared models according to the Akaike Information Criterion (AIC). We used the absolute abundance values (bacterial counts normalized per unit of ant biomass) as response variable, and bacterial strain (i.e. *EntAcro1* or *EntAcro10*), phylogenetic host group (i.e. leaf-cutting or non-leaf-cutting ants), and the statistical interaction between these predictors as fixed categorical variables. We evaluated significant differences across groups using post-hoc comparisons

as planned contrasts and Bonferroni corrections based on the 'glht' function in the 'multcomp' package (*Hothorn et al., 2008*).

## Collection of forage substrates harvested by colonies in the field

Foraging substrate preference data were collected in the field from nine attine ant species (*T. cornetzi*, *T. zeteki*, *S. amabilis*, *C. costatus*, *C. longiscapus*, *C. rimosus*, *Myc. smithii*, *Myr. ednaella* and *Ap. dentigerum*) in Gamboa, Panama and represent 101 hr of observation time on 103 colonies (*T. cornetzi* (n = 48), *T. zeteki* (n = 12), *S. amabilis* (n = 6), *C. costatus* (n = 8), *C. longiscapus* (n = 5), *C. rimosus* (n = 8), *Myc. Smithii* (n = 14) and *Ap. dentigerum* (n = 2). Colonies were located and marked in several field sites within lowland Panamanian rainforest near Gamboa after placing polenta baits in the leaf litter and then tracking workers back to their nests when vouchers of workers were collected in EtOH to allow identification. After at least a week, laden returning foragers were observed with a headlamp during set observation periods. Colonies were typically observed during 60 min intervals (59 ± 14 min), although observations were cut short in the case of rain, or extended in the case of very slow foraging (*e.g.* 120 min intervals for *Ap. dentigerum*). Harvested substrates were carefully removed from the mandibles of the workers, collected in eppendorf tubes, and returned to the lab where they were dried at 60℃ for 24 h and then sorted under a dissecting microscope and catalogued. Substrates collected by the ants were split in seven categories (leaves, fruits, flowers, seeds, wood fragments, insect frass and 'other', that is, small dead insect fragments, pieces of unidentifiable detritus, and putative bird feces) for which counts per observed colony were generated. Similar data were extracted from a recently published study conducted at the same field site at the same time of year (May) but focusing on six leaf-cutting ant species (*At. colombica*, *At. cephalotes*, *At. sexdens*, *Ac. echinatior*, *Ac. octospinosus* and *A. volcanus*) in Gamboa, Panama (*Kooij et al., 2014a*). Merged datasets were normalized by converting all observations to total observed counts in one hour.

To visualize foraging substrate preferences across the attine ant species, we converted the count data to proportions and performed an unscaled Principal-Component Analysis (PCA) in R using the 'ade4' package. To independently verify the statistical significances obtained, we used the mean of the replicated count values per colony. We then fitted the normalized hourly count data in a zero-inflated negative binomial (ZINB) regression model with the function 'zeroinfl' in the package 'pscl' (*Kleiber et al., 2008*). ZINB regression is typically used for true count variables to model positively-skewed data with an abundance of zeros and it fitted our data better than a zero-inflated Poisson or a negative binomial generalized linear model (GLM) without zero-inflation when we compared them using the Akaike Information Criterion (AIC) and the Vuong's non-nested test ('vuong' function in 'pscl' package). We used the absolute itemized foraging substrate preference values as response variable and the interaction of the foraging substrate types (i.e. leaves, fruits, flowers, seeds, insect frass and other) and phylogenetic group (i.e. leaf-cutting *versus* non-leaf-cutting ants) as fixed categorical variables. We conducted Tukey's HSD post hoc tests for each substrate type between leaf-cutting and non-leaf-cutting using the 'lsmeans' package (*Lenth, 2016*).

We used Mantel tests to compare differences in absolute *EntAcro1* or *EntAcro10* abundances (calculated for each species using the qPCR data) with the overall dissimilarities in their foraging substrate preferences. We used only data from the 12 attine ant species which were common in both datasets (*At. colombica*, *At. cephalotes*, *At. sexdens*, *Ac. echinatior*, *Ac. octospinosus*, *T. cornetzi*, *T. zeteki*, *S. amabilis*, *C. costatus*, *C. rimosus*, *Myc. smithii* and *Ap. dentigerum*). We created a bray-curtis distance matrix for each of the *EntAcro* strains and a similar dissimilarity matrix based on the the seven foraging substrate categories using bray-curtis distance in R. The final Mantel tests were performed in R using 10,000 permutations.

## Acknowledgements

We thank Panagiotis Ioannidis, Luigi Pontieri, David Nash and Lucas Schrader for bioinformatical and statistical advice, Sen Li for allowing access to the server that ran all phylogenomic analyses, Anna Fomsgaard for helping with the RNA extractions and cDNA synthesis of ant colonies, David Donoso for verifying the species identification of our vouchers of attine ant workers, and Ernesto Gomez for assistance in the field. The Smithsonian Tropical Research Institute in Panama made facilities available, the Autoridad Nacional del Ambiente (ANAM) of Panama issued collection and export permits,

and Rachelle Adams allowed us to use colony RMMA150520-03. Funding was provided by the Danish National Research Foundation (DNRF57), an ERC Advanced grant (323085) to JJB, and Marie Sklodowska-Curie Fellowships to PS (300584), MZ (660255) and JZS (327940), who was also supported by a Postdoctoral Fellowship from the Smithsonian Institution Competitive Grants Program (to WT Wcislo, JJB, and JZS).

## Additional information

### Funding

| Funder | Grant reference number | Author |
|---|---|---|
| Danmarks Grundforsknings-fond | DNRF57 | JJ Boomsma |
| European Research Council | ERC Advanced Grant 323085 | JJ Boomsma |
| H2020 Marie Skłodowska-Curie Actions | IEF 300584 | Panagiotis Sapountzis |
| H2020 Marie Skłodowska-Curie Actions | IIF 327940 | Jonathan Z Shik |
| Smithsonian Institution | Postdoctoral fellowship | Jonathan Z Shik |
| H2020 Marie Skłodowska-Curie Actions | IEF 660255 | Mariya Zhukova |

The funders had no role in study design, data collection and interpretation, or the decision to submit the work for publication.

### Author contributions

Panagiotis Sapountzis, Conceptualization, Data curation, Formal analysis, Funding acquisition, Validation, Investigation, Methodology, Writing—original draft, Writing—review and editing; Mariya Zhukova, Validation, Methodology; Jonathan Z Shik, Resources, Data curation; Morten Schiott, Supervision, Validation; Jacobus J Boomsma, Conceptualization, Resources, Supervision, Funding acquisition, Validation, Writing—original draft, Project administration, Writing—review and editing

### Author ORCIDs

Panagiotis Sapountzis http://orcid.org/0000-0001-6286-3918
Jacobus J Boomsma https://orcid.org/0000-0002-3598-1609

### Decision letter and Author response

Decision letter https://doi.org/10.7554/eLife.39209.036
Author response https://doi.org/10.7554/eLife.39209.037

## Additional files

### Supplementary files

• Supplementary file 1. Summary statistics of the metagenomic sequencing of the endosymbiotic Mollicutes *EntAcro1* and *EntAcro10*. Following the assembly, contigs were separated into phylogenetic bins, which are presented in each column. (**A**) Sequencing statistics, annotation and completeness assessment using the BUSCO orthologs. (**B**) Total numbers of predicted coding sequences (CDS) that gave top matches with different bacterial classes for *EntAcro1*. (**C**) Comparable data for the *EntAcro10* bins. In both data sets, 'A' bins were the ones that contained contigs with clear similarities to Mollicutes, while 'B' bins mostly represented sparse contaminants, and the 'C' bin (only *EntAcro1*) contained sequences that belonged to bacteriophages of the family *Microviridae* (see text for details). Following the results of the BUSCO analysis, we examined the complete genome sequence of the *M. florum* symbiont associated with plants (*Baby et al., 2013*) using the same

software, which gave the same numbers of genes as for *EntAcro1*. These results suggested that our 'A' bins represent proper draft genomes of single *EntAcro1* and *EntAcro10* strains.

DOI: https://doi.org/10.7554/eLife.39209.024

• Supplementary file 2. The 65 single-copy orthologs used in our phylogenetic reconstructrions as identified by Orthofinder based on 59 Mollicutes genomes, associated mainly with mammal, insect and crustacean hosts (see Materials and methods). Genome names are presented at the top of each column with the same identifiers as in the ensembly database (http://bacteria.ensembl.org/info/website/ftp/index.html). NCBI accession numbers are given for all ortholog genes (rows) except for EntAcro1 and EntAcro10 where we used the RAST annotation identifiers.

DOI: https://doi.org/10.7554/eLife.39209.025

• Supplementary file 3. Differences in predicted coding sequences of the fungus-growing ant symbionts EntAcro1A and EntAcro10A and the genomes of closely related *Spiroplasma* strains associated with mosquito, honeybee and fly hosts that also have specialized diets. The main table (**A**) shows the predicted set of genes always present in EntAcro1A while at the same time overlapping with EntAcro10A (the 39 top rows) and to a variable extent also with the twelve other *Spiroplasma* strains (73 bottom rows). KEGG annotation, eggNOG functional and hierarchical categories, subcategories, subsystems and roles for the predicted genes, as defined by the RAST annotation server, are presented as columns while rows show presences and absences. EntAcro1A genes were sorted first for being shared or not with EntAcro10A, then with the mosquito-associated *S. sabaudiense*, *S. diminutum*, *S. culicicola*, *S. taiwanense*, then with the bee-associated *S. melliferum* and *S. apis*, and finally with the fly-associated *S. atrichopogonis*, *S. chrysopicola*, *S. syrphidicola*, and *S. poulsonii*. Presences of genes in strains documented or believed to be pathogenic are given in pink and presences in putatively mutualistic or commensal strains are given in light blue (same colors as in *Figures 1* and *5*). For strains without clear literature references indicating pathogenicity (implying they are unlikely to be pathogens; *Supplementary file 4*) we used a dark gray font. Enriched complete pathways of *EntAcro1* are highlighted with different background colors across rows, using gray and black for complete pathways that are common in bacteria and usually have housekeeping functions or are involved in homologous recombination and are thus not of primary interest. Complete pathways that were of focal interest (based on the eggNOG annotation) are marked in bright colors: shades of green for those involved in metabolism, red for those involved in information processing, and blue for those related to cellular function (same as in *Figure 1—figure supplement 1*). The legend directly below the table provides short descriptions for each of the highlighted pathways. The separate table (**B**) summarizes the gene-distributions for the pathways described in Table A, with regard to: (i) pathogenic and non-pathogenic associations of Mollicutes strains, (ii) the respective hosts strains they are associated with, and (iii) bacterial phylogeny (*Figure 1*).

DOI: https://doi.org/10.7554/eLife.39209.026

• Supplementary file 4. Symbiotic associations of the Mollicutes strains for which sequenced genomes were available for phylogenetic comparison with the focal *EntAcro1* and *EntAcro10* symbionts of the present study. From left to right: the 59 Mollicutes strain IDs for which a phylogeny was constructed (*Figure 1*, *Figure 1—figure supplement 2*) using the same identifiers as in the Ensembl database (http://www.ensembl.org), their most likely symbiotic associations (based on the available literature), their hosts, and previous studies that examined their symbiotic interactions.

DOI: https://doi.org/10.7554/eLife.39209.027

• Supplementary file 5. Primers used in the present study. From left to right: Primer names, the function of the gene targeted based on the RAST annotation, the target organism, the primer sequences from 5' to 3' and the annealing temperatures used for the qPCR.

DOI: https://doi.org/10.7554/eLife.39209.028

• Supplementary file 6. Correlation statistics for the association between the number of EntAcro1 cells and the expression of 10 MCT-like genes in the *Ac. echinatior* midgut and fat body tissues. From left to right we present for each gene examined: The protein ID in the Uniprot database (http://www.uniprot.org/), the gene name and the gene ID based on the *Ac. echinatior* annotation, the predicted protein size, and the results of statistical analyses examining the correlation between expression of the bacterial ftsZ gene in midgut/fat body tissues and the expression of MCT-like genes. The significance of each correlation was first evaluated using a non-parametric Spearman correlation test (see Methods), for which we give ρ values, p values and Bonferroni corrected p values.

The only significant correlation (marked in bold) was between the ftsZ and the MCT1 (F4WHWZ) transporter, for which previous functional experiments have demonstrated that it is actively involved in the import of acetate in eukaryotic cells (*Kirat and Kato, 2006*; *Moschen et al., 2012*).

DOI: https://doi.org/10.7554/eLife.39209.029

• Transparent reporting form

DOI: https://doi.org/10.7554/eLife.39209.030

### Data availability

Sequencing data were deposited in the NCBI Genome submission portal under accession numbers SAMN06251630 and SAMN06251631.

The following datasets were generated:

| Author(s) | Year | Dataset title | Dataset URL | Database and Identifier |
|---|---|---|---|---|
| Panagiotis Sapount- zis | 2015 | Entomoplasmatales bacterium EntAcro1 | https://www.ncbi.nlm. nih.gov/biosample/? term=SAMN06251630 | NCBI BioSample, SAMN06251630 |
| Sapountzis P | 2015 | Entomoplasmatales bacterium EntAcro10 | https://www.ncbi.nlm. nih.gov/biosample/? term=SAMN06251631 | NCBI BioSample, SAMN06251631 |

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
