## [Decision Letter]

Thank you for submitting your article "Reconstructing the functions of endosymbiotic Mollicutes in fungus-growing ants" for consideration by *eLife*. Your article has been reviewed by three peer reviewers, one of whom is a member of our Board of Reviewing Editors, and the evaluation has been overseen by Ian Baldwin as the Senior Editor. The following individual involved in review of your submission has agreed to reveal his identity: Martin Kaltenpoth (Reviewer #3).

The reviewers have discussed the reviews with one another and the Reviewing Editor has drafted this decision to help you prepare a revised submission.

Summary:

Sapountzis and colleagues report on their recent comparative genomic and experimental analysis of two Mollicutes symbionts that are widespread among leaf-cutter ants and are essential for the three-partite interaction between leaf-cutter ants, their fungal cultivar and bacterial symbionts. What makes this paper valuable is the combination of solid genomic analyses of unculturable symbionts with manipulative experimentation that directly implicate the symbionts in the metabolism of the host. The manuscript is well written. First, two draft genomes of very distinct and commonly present strains are reported and analyzed with respect to certain nutritional and defensive function supporting the symbiosis. They provide evidence for the utilization of glucose, arginine, citrate, and N-acetyl-glucosamine by the symbionts, and particularly arginine metabolism.

However, two of three reviewers feel that the conclusions made within this manuscript are in part highly speculative and the data not strong enough to support the case as presented.

Essential revisions:

The following points have been raised and need to be addressed within the manuscript:

1) Draft genomes are by nature incomplete and are particularly difficult to interpret unambiguously. There is a potential issue with missing metabolic genes in the genome. Even though the genomes have been sequenced to a high degree of completeness, a substantial portion of the genes encode poorly characterized proteins (see Figure 1—figure supplement 1 legend; 338 and 358 genes for the two symbionts, respectively). Can any information be derived on these genes by consulting additional databases or characterizing functional domains? Otherwise, a considerable fraction of the symbionts' metabolism may be missing. While this is not unusual for the reconstruction of bacterial metabolism, the number of uncharacterized proteins is very high for a symbiotic bacterium with such a small genome. Is this the same for other Mollicutes associated with different hosts?

2) Various elements are hard to interpret – identifying transporters as transporters is commonly straightforward, but identifying what is transported through them (very important in this manuscript) is very hard – similarity to database hits is not a very strong indicator, as they evolve quite freely to new functions (and the database hits are quite a long way away).

3) The authors use the term functional genomics (Introduction, fifth paragraph), but do so wrongly: functional genomics is taking genome sequences and performing experimental analysis of function through knock out mutations etc.; what the authors are doing is comparative genomics, where they are inferring properties by comparison – an altogether different and less inferentially powerful approach (albeit where one would start any process) as it relies completely on homology, and ignores the properties of uncharacterized genes.

4) All three reviewers agreed that the interpretation of the CRISPR-CAS and mutualism needs major revision. The presence of the system is interesting, because most obligate symbionts do not have this system. Presumably, this is because most obligate symbionts are endosymbiotic, and thus environmental phage exposure is rare. As these symbionts are not endosymbiotic, the evolutionary argumentation made is highly speculative.

However, it seems likely that the environment and the associated risk of being attacked by a phage plays a much more important role for the cost/benefit ratio of maintaining the defense than the effect of the microbe on the host. Inferring here that the high amount of resources provided by the host is beneficial for the host-symbiont alliance because it allows for maintaining the defenses in the symbiont seems very far-fetched. It is equally plausible to assume that a pathogen would find a rich environment in the gut, but also abundant phages that it needs to defend itself against. As with the opportunity for HGT, it would be better to discuss the findings in the context of the symbionts' localization (i.e. gut vs. more specialized structures or within host cells), rather than pathogenicity vs. mutualism. If other Mollicutes occur within host cells or in specialized organs, the risk of infection with phages can be expected to be much lower, which could explain the loss of defense systems.

5) With respect to physiological experiments, these rely on comparison of native microbiome versus antibiotic treated. These types of experiments have limited power in the form presented, as the antibiotic treatment alters many aspects of host biology – not just the microbes targeted. It alters other gut microbiome components, and more importantly, tetracycline and rifampicin additionally have non-target effects on eukaryotic cells (tetracycline in particular damages mitochondria). Thus this type of experiment does not allow a strong causal inference that host changes are associated with particular microbes – as the treatment effect go beyond those particular microbes. With culturable microbes, the 'elegant' experiment is to treat and reintroduce (equalizing antibiotic impact between classes). In conclusion, the inferential value of experiments is reduced by the inevitable correlated effects of antibiotics. In summary, the authors should be more careful with their discussion and data interpretation in this section.

6) The authors should be more careful with the word coevolved for these symbiosis until both host and microbe adaptations to each other (and indeed adaptation of one producing adaptation of the other) is shown. They are likely nevertheless to be co-adapted.

---

## [Author Response]

Essential revisions:The following points have been raised and need to be addressed within the manuscript:1) Draft genomes are by nature incomplete and are particularly difficult to interpret unambiguously. There is a potential issue with missing metabolic genes in the genome. Even though the genomes have been sequenced to a high degree of completeness, a substantial portion of the genes encode poorly characterized proteins (see Figure 1—figure supplement 1 legend; 338 and 358 genes for the two symbionts, respectively). Can any information be derived on these genes by consulting additional databases or characterizing functional domains? Otherwise, a considerable fraction of the symbionts' metabolism may be missing. While this is not unusual for the reconstruction of bacterial metabolism, the number of uncharacterized proteins is very high for a symbiotic bacterium with such a small genome. Is this the same for other Mollicutes associated with different hosts?

We appreciate this comment and have now done specific probing of both our own annotations and those of the published insect-associated Mollicutes genomes. The results, presented in a new table (Figure 1—figure supplement 3—source data 1) show that all 14 Mollicutes examined so far have similarly high fractions of genes (18-25%) with unknown functions as our attine symbionts *EntAcro1* and *EntAcro10*. After now documenting this general pattern, the next question is whether percentages of genes with unknown functions have any bearing on the likelihood of a bacterium to be a mutualistic symbiont. It is true that some obligate (mainly intracellular) symbionts like *Buchnera* have lower (ca 10%) percentages of genes with unknown function, but these bacteria have extremely reduced genomes due to selection for maintaining only genes relevant to mutualistic services to the host (Toft and Andersson 2010; McCutcheon and Moran 2012). The *EntAcro1* and *EntAcro10* symbionts that we studied do not belong to this ‘*Buchnera*-like’ category of ancient organelle-line endosymbionts that evolved hundreds of millions of years ago, but rather to a more recent category of symbionts that have a history of some tens of millions of years of co-adaptation with hosts that led to functional complementarity in varying degrees but without extreme degrees of genome reduction. In our revision, we have now taken care to make these points explicit: 1. That several Mollicutes associated with insect hosts have similar percentages of genes with unknown function (Figure 1—figure supplement 1) consistent with none of them having co-evolved long enough with their hosts to have obtained the stably-reduced non-recombining genomes that typically have mostly well characterized symbiotic genes (e.g. metabolic complementarity related genes), and 2. That our inferred dates of origin of the associations of *EntAcro1* and *EntAcro10* with leafcutting and higher attine ants (ca. <20 MYA and 25 MYA, respectively) gave us no reason to expect extreme degrees of genome reduction, consistent with what we found (subsection ““The arginine recycling niche of Spiroplasma-like abdominal symbionts“; based on available phylogeny data on fungus-growing ants by Schultz and Brady 2008; Nygaard et al., 2016; Branstetter et al., 2017).

2) Various elements are hard to interpret – identifying transporters as transporters is commonly straightforward, but identifying what is transported through them (very important in this manuscript) is very hard – similarity to database hits is not a very strong indicator, as they evolve quite freely to new functions (and the database hits are quite a long way away).

This is a reasonable critique. We feel we have done the best possible job in exploiting the online data available, but we agree that the present stage of knowledge can only produce likelihood inferences of the metabolites being transported. We have now gone through the manuscript text and rephrased our wording to reflect that our inferences will require precise follow-up studies (e.g. Results section; Figures 2 and 4).

3) The authors use the term functional genomics (Introduction, fifth paragraph), but do so wrongly: functional genomics is taking genome sequences and performing experimental analysis of function through knock out mutations etc.; what the authors are doing is comparative genomics, where they are inferring properties by comparison – an altogether different and less inferentially powerful approach (albeit where one would start any process) as it relies completely on homology, and ignores the properties of uncharacterized genes.

We have come across a sufficient number of recent papers using functional genomics in the same loose sense as we did to feel this is more a semantic issue than phrasing that is right or wrong in an absolute sense. This is also why we only mentioned the term once and we did not imply to present a functional genomics study ourselves – we only stated that such studies are missing from the field. However, we appreciate that it is important to be precise, so in the revision we have now removed the use of the term ‘functional genomics’ altogether.

4) All three reviewers agreed that the interpretation of the CRISPR-CAS and mutualism needs major revision. The presence of the system is interesting, because most obligate symbionts do not have this system. Presumably, this is because most obligate symbionts are endosymbiotic, and thus environmental phage exposure is rare. As these symbionts are not endosymbiotic, the evolutionary argumentation made is highly speculative.However, it seems likely that the environment and the associated risk of being attacked by a phage plays a much more important role for the cost/benefit ratio of maintaining the defense than the effect of the microbe on the host. Inferring here that the high amount of resources provided by the host is beneficial for the host-symbiont alliance because it allows for maintaining the defenses in the symbiont seems very far-fetched. It is equally plausible to assume that a pathogen would find a rich environment in the gut, but also abundant phages that it needs to defend itself against. As with the opportunity for HGT, it would be better to discuss the findings in the context of the symbionts' localization (i.e. gut vs. more specialized structures or within host cells), rather than pathogenicity vs. mutualism. If other Mollicutes occur within host cells or in specialized organs, the risk of infection with phages can be expected to be much lower, which could explain the loss of defense systems.

We appreciate this feedback. It has three components which we address below:

A) Whether *EntAcro1* and *EntAcro10* are endosymbionts or not. B) Whether our hypothetical cost/benefit interpretation of the maintenance of multiple defense systems is reasonable relative to alternative interpretations, and C) Whether *EntAcro1* and *EntAcro10* are unique among the insect-associated Mollicutes in being extracellular symbionts and whether that compromises our hypothetical interpretation of the evolutionary cost/benefits of maintaining these defenses.

A) Whether *EntAcro1* and *EntAcro10* are endosymbionts or not appears to be largely a semantic issue. Some researchers define an endosymbiont as living exclusively within the confinements of a eukaryotic cell (similar to mitochondria and plastids), but that restricted definition is not generally agreed upon and is unnecessary restrictive for symbionts with a shorter evolutionary history and less reduced genomes because they maintain a wider set of niches inside the body of insect hosts. A series of specialized gut symbionts, as for example the gut bacteria of social bees (Kwong and Moran Nat. Rev. Microbiol. 2016) are a case in point. They are endosymbionts with clear nutrient supplementation and disease-protection functions even though they are associated with the gut lumen. Such symbionts fit the broader and we believe classic definition (Bourtzis and Miller 1998 CRC press; Insect Symbiosis) that defines an endosymbiont as living inside the body of another organism regardless of being localized intra- or extracellularly. The opposite of this definition is an ectosymbiont, which lives on the outside of a host – cuticular Actinobacteria in beewolves and attine ants are a good example. We have now made this definition explicit at the start of our revised manuscript (Introduction, first paragraph), and have endeavored to use the terms ‘intracellular endosymbiont’ and ‘extracellular endosymbiont’ whenever there might otherwise arise ambiguity (Introduction, last paragraph and subsection “Substrate utilization and reconstruction of metabolic pathways”, last paragraph). We feel precision of terms is important because an increasing number of symbionts that were previously believed to be strictly intracellular endosymbionts have now been shown to occur both extracellularly and intracellularly (e.g. Wolbachia; cf Andersen et al., 2012).

B) Whether our hypothetical cost/benefit interpretation of the maintenance of multiple defense systems is reasonable relative to alternative interpretations can indeed be debated, but we maintain it is more likely than assumed by reviewers. First, we would like to point out that we present the connection between symbiotic status (actively maintained mutualist versus actively antagonized pathogen) and the presence/absence of phage defense systems as an ultimate causation hypothesis, that is, as a hypothesis of what would make sense as the outcome of a co-adaptation process on a scale of some millions of years. As Nobel Laureate Niko Tinbergen made clear in his classic paper (1963), such ultimate questions are different from, and complementary to, the proximate (molecular, HGT and gene expression) mechanisms that would be needed to produce such adaptive interactions. We agree that we should have explicitly addressed the issue of tissue localization (which we now do in the revised manuscript; “The costs and benefits of defending domesticated bacterial symbionts in the gut”), because it actually strengthens the logic of our argument. We believe that our evolutionary hypothesis is plausible when applied to the gut lumen symbionts of attine ants because:

a) It is well documented that bacterial defense systems have a cost and that they should be rather quickly lost or become inactive when such systems are ‘unemployed’ by lack of phage challenges, because the costs of maintaining these defense functions would then impose a clear fitness disadvantage (e.g. Vasu and Nagaraja 2013; Vale et al., 2015; Burstein et al., 2016).

b) Our study shows that *EntAcro1* and *EntAcro10* can both live intracellularly in the fat body and extracellularly in the gut lumen (Sapountzis et al., 2015). We agree with the reviewers that exposure to phage attack is less likely for intracellular symbionts, but our argument is about the extracellular Mollicutes that the hosts appear to farm in the gut lumen on demand (i.e. allow to multiply when there is plenty of arginine or citrate and to wither when there is little). Such hypothesized host regulation of extracellular bacterial titers in response to specific needs for arginine recycling in the hindgut and citrate catabolism in the midgut (Figure 4—figure supplement 1) would make the symbiosis with Mollicutes highly beneficial and is consistent with our data so far. In the revised text we have now made explicit that this is a hypothetical scenario that matches our data and would be a meaningful outcome of a co-adaptation process, and that is also consistent with other studies that suggested/demonstrated host control over symbiont titers of intracellular symbionts (e.g. Wilkinson et al., AEM 2006; Russell et al., 2014).

c) A co-adapted mutualistic symbiont in the gut lumen will thus be actively encouraged by host resources to reach periodically high densities in a way that would never happen to a commensal/transient microbe or a pathogenic bacterium in a healthy host because the remaining gut microbiota have been selected to competitively suppress that kind of useless or damaging microbes – we believe this kind of dynamics is also well supported for a series of gut microbiota of humans, mammals and invertebrates (e.g. Baumler and Sperandio Nature 2016).

d) Assuming this reasoning is correct and knowing that high bacterial densities disproportionally increase the risks of phage attack and the likelihood of bacterial defense mechanisms to be maintained by selection (e.g. Thingstad et al., PNAS, 2014), it follows logically that beneficial gut symbionts that are actively maintained (resource-provisioned) by the host to occasionally reach high densities are expected to have phage defense systems that co-occurring commensal-transient/pathogenic bacteria should not have been selected to maintain.

e) We admit that an expectation of presence of phage-defense systems does not prove that such defenses would always be acquired. There may be constraints on what HGT can establish and in the predictability of positive selection for maintaining complete defense systems. Hence it is perhaps not surprising that other insect hosts with Mollicutes endosymbionts show at best signs of partial maintenance of phage-defense systems, and this may well be because we lack positive evidence that these other bacteria are in the gut lumen of flies and bees (rather than only intracellularly) and that the titers in these systems are as highly variable as in the attine ants. However, the consistent presence of two phage defense systems in both *EntAcro1* and *EntAcro10* strains needs an explanation and as far as we can see there is no satisfactory alternative evolutionary explanation. This is because our scenario is actually consistent with established evolutionary theory on levels-of-selection, transmission efficiency, and the expression of competitive symbiont traits for which we have now added a few references in the final Discussion paragraph. In such a situation, we believe that modest informed speculation is reasonable because it will encourage further research.

f) Our hypothetical scenario actually predicts the kinds of endosymbiotic mutualists that should possess functional phage defense systems, that is, we expect these only in clearly beneficial gut symbionts and not in similarly beneficial intracellular endosymbionts when both are vertically transmitted. We should also only expect consistent selection for maintaining phage-defenses in mutualistic symbionts that periodically or permanently occur in high densities while occurring in organs that are not closed compartments. Also gut symbionts of honeybees have been shown to carry bacteriophage defense systems (Kwong et al., PNAs 2014) and we just discovered that a Rhizobiales symbiont from the hindgut lumen of *Acromyrmex* leafcutter ants, whose genome sequences we are currently analyzing, also has a CRISPR and a Restriction-Modification system (Zhukova et al., in prep). Finally, the evolutionary logic depends on whether symbionts are vertically or horizontally transmitted, which remained implicit in our previous discussion. We now make that explicit (and link it to a few new theoretical references) and explain that gut symbionts in ant colonies are actually vertically transmitted by default because dispersing genes are inoculated by sister workers.

In light of the arguments above we have maintained our hypothetical interpretations, but significantly edited the presentation to make sure we: 1) Emphasize this is an ultimate evolutionary cost/benefit hypothesis that remains to be further tested for its general applicability, that may not always be valid, and that will not be understood until all proximate mechanisms (e.g. tissue localization, chromosomal stability) have been thoroughly investigated and clarified as well, 2) Make explicit that our hypothesis is specific for gut-inhabiting symbionts at high or highly-fluctuating densities that are transmitted only to ants of the same colony and to virgin dispersing queens. 3) Make explicit that no such co-adaptation would arise over night, but over sufficient evolutionary time to allow hosts to evolve reasonable control over the symbiont densities, 4) Emphasize that we present this scenario at some length to raise awareness of this interesting possible type of co-evolutionary dynamics consistent with evolutionary theory, well knowing that explicit tests are needed to substantiate these claims (subsection “The costs and benefits of defending domesticated bacterial symbionts in the gut”, second paragraph).

We hope the reviewers will find these revisions appropriate. If reservations would persist we would be happy to see a comment posted by one or several of the reviewers, as that will likely stimulate the further empirical work needed to test the validity of our combined inferences.

5) With respect to physiological experiments, these rely on comparison of native microbiome versus antibiotic treated. These types of experiments have limited power in the form presented, as the antibiotic treatment alters many aspects of host biology – not just the microbes targeted. It alters other gut microbiome components, and more importantly, tetracycline and rifampicin additionally have non-target effects on eukaryotic cells (tetracycline in particular damages mitochondria). Thus this type of experiment does not allow a strong causal inference that host changes are associated with particular microbes – as the treatment effect go beyond those particular microbes. With culturable microbes, the 'elegant' experiment is to treat and reintroduce (equalizing antibiotic impact between classes). In conclusion, the inferential value of experiments is reduced by the inevitable correlated effects of antibiotics. In summary, the authors should be more careful with their discussion and data interpretation in this section.

We agree and share the notion that our artificial diets in the antibiotics experiment may have induced confounding effects. We have now added an explicit caveat on this in line with the comment above (subsection “Resource acquisition, gene expression and inferred Mollicutes functions”, second paragraph). However, we also note that the antibiotics treatment was not the only evidence, because we also had an additional treatment with reduced number of Mollicutes but without antibiotics. The fact that the antibiotics treatment linearly extended the trend observed between the additional treatment and control colonies suggests that the confounding effect has not been major.

6) The authors should be more careful with the word coevolved for these symbiosis until both host and microbe adaptations to each other (and indeed adaptation of one producing adaptation of the other) is shown. They are likely nevertheless to be co-adapted.

We agree and have gone through the manuscript to examine instances where we used the word ‘coevolved’. Only one was in fact found (in the Discussion) which has now been replaced by ‘co-adapted’.